# Alterations of HDL’s to piHDL’s Proteome in Patients with Chronic Inflammatory Diseases, and HDL-Targeted Therapies

**DOI:** 10.3390/ph15101278

**Published:** 2022-10-18

**Authors:** Veronika Vyletelová, Mária Nováková, Ľudmila Pašková

**Affiliations:** Department of Cell and Molecular Biology of Drugs, Faculty of Pharmacy, Comenius University, 83232 Bratislava, Slovakia

**Keywords:** HDL, chronic low-grade inflammation, atherosclerosis, piHDL, HDL therapy

## Abstract

Chronic inflammatory diseases, such as rheumatoid arthritis, steatohepatitis, periodontitis, chronic kidney disease, and others are associated with an increased risk of atherosclerotic cardiovascular disease, which persists even after accounting for traditional cardiac risk factors. The common factor linking these diseases to accelerated atherosclerosis is chronic systemic low-grade inflammation triggering changes in lipoprotein structure and metabolism. HDL, an independent marker of cardiovascular risk, is a lipoprotein particle with numerous important anti-atherogenic properties. Besides the essential role in reverse cholesterol transport, HDL possesses antioxidative, anti-inflammatory, antiapoptotic, and antithrombotic properties. Inflammation and inflammation-associated pathologies can cause modifications in HDL’s proteome and lipidome, transforming HDL from atheroprotective into a pro-atherosclerotic lipoprotein. Therefore, a simple increase in HDL concentration in patients with inflammatory diseases has not led to the desired anti-atherogenic outcome. In this review, the functions of individual protein components of HDL, rendering them either anti-inflammatory or pro-inflammatory are described in detail. Alterations of HDL proteome (such as replacing atheroprotective proteins by pro-inflammatory proteins, or posttranslational modifications) in patients with chronic inflammatory diseases and their impact on cardiovascular health are discussed. Finally, molecular, and clinical aspects of HDL-targeted therapies, including those used in therapeutical practice, drugs in clinical trials, and experimental drugs are comprehensively summarised.

## 1. Introduction

Patients with chronic inflammatory diseases, such as rheumatoid arthritis (RA), systemic lupus erythematosus (SLE), diabetes mellitus (DM), periodontitis (PD), non-alcoholic steatohepatitis (NASH), non-alcoholic fatty liver disease (NAFLD) or chronic kidney disease (CKD) are at increased risk of premature cardiovascular disease (CVD)-associated mortality in comparison to the general population [1,2,3,4]. This increased CVD morbidity and mortality may not be attributed only to classical general known CVD risk factors such as obesity, hyperlipidaemia, smoking, hypertension, or lack of physical activity. Inflammatory processes in various tissues, such as the liver, joints, or intestine, can lead to systemic inflammation, which may aggravate or even initiate comorbid pathologies [1,5,6,7,8]. A common virtue of chronic inflammatory diseases is low-grade systemic inflammation, usually characterised by increased pro-inflammatory mediators like interleukin-1 (IL-1), IL-6, C-reactive protein (CRP), and others [7,9]. Despite the heterogeneity in the designs, populations, and methods of analysis of the various clinical studies, it can be assumed that chronic systemic inflammation is the main contributor to accelerated atherosclerosis. Atherosclerosis, an inflammatory disease developing in the arterial wall, is referred to be the main cause of coronary artery disease (CAD) and belongs to the leading causes of morbidity and mortality worldwide.

A low concentration of high-density lipoprotein (HDLc) was identified as an independent predictor of CVD risk [10]. However, epidemiologic data from patients with chronic inflammatory diseases did not support the improvement of clinical parameters following the increase in HDLc, indicating that high-density lipoprotein (HDL) quantity does not predict CVD risk under inflammatory conditions. It was discovered that depending on the presence or absence of inflammation, HDL may change its functionality and act both anti-inflammatory and pro-inflammatory. Thus, the determination of HDL functionality could be a better predictor of CVD risk instead of the standard estimation of serum lipid levels [11,12,13,14]. Functional anti-inflammatory HDL contains a high number of active proteins and enzymes with atheroprotective activities, including cholesterol acceptor function, antioxidative, anti-inflammatory, antiapoptotic, antiplatelet, and antithrombotic effects. Under inflammatory conditions, alterations in HDL’s proteome and lipidome may occur, enabling HDL to act as an important component of innate immunity. Such pro-inflammatory HDL (piHDL) possesses the ability to enhance inflammatory answer to infection by changes in cholesterol homeostasis, direct binding to pathogens, and exerting immunomodulatory functions in macrophages and endothelial cells (ECs) [15,16]. Although these pro-inflammatory alterations of HDL work protective in the case of acute infection, chronic systemic inflammation accompanied by oxidative stress triggers long-term alterations in HDL function, which can lead to accelerated atherosclerosis [17,18,19].

During the shift from HDL to piHDL, a complex of dynamic changes between interacting proteins and lipids occurs. These processes likely include oxidation of lipids and lipoproteins in HDL, e.g., as a consequence of the increased activity of peroxidases, decreased synthesis of proteins, exchanging of proteins participating in reverse cholesterol transport (RCT), and antioxidative enzymes for pro-oxidative proteins [9,20]. piHDL is depleted of apolipoprotein A-I (apoA-I), paraoxonase 1 (PON1), and other components. On the other hand, piHDL is enriched in oxidated phospholipids and lysophospholipids, free cholesterol, free fatty acids (FFAs), and triacylglycerols (TAGs), as well as pro-inflammatory proteins, such as serum amyloid A (SAA) and ceruloplasmin. These components endorse oxidation of low-density lipoprotein (LDL), endothelial dysfunction and induce the chemotactic activity of monocytes and secretion of pro-inflammatory molecules. Importantly, piHDL partly loses RCT function, hence the HDL-mediated cholesterol efflux is decreased. Consequently, the process of atherosclerosis accelerates [11,12,13,14]. The presence of dysfunctional piHDL was observed in numerous pathological processes associated with systemic inflammation, such as coronary heart disease (CHD), DM, CKD, chronic infections, and some rheumatoid diseases [11,21]. Since the first reference about the existence of pro-inflammatory HDL nature [22], numerous clinical and experimental studies in animal models and *in vitro* cell cultures were performed to explain the molecular mechanisms of disfavourable HDL shift and its consequences [23,24,25,26]. Identification of protein profiles that are associated with anti-inflammatory HDL versus piHDL, and their consequences for HDL’s functionality, could lead to the development of new strategies for early detection and therapeutic interventions in atherosclerosis.

## 2. Chronic Inflammatory Diseases Are Associated with Increased CVD Risk

Inflammation, lipid, and glucose metabolism, and atherosclerosis are strongly interconnected. They may influence each other, share common pathways, and together they create a very complex system. This interconnection is so intense, that it is commonly termed as, immunometabolism”. Inflammatory cytokines regulate several metabolic pathways [27], on the other hand, cholesterol [28] or FFAs [29] could provoke inflammatory processes. In addition, dysregulated glucose metabolism in diabetic patients is often accompanied by lipid abnormalities and a pro-atherogenic environment. This interplay may explain why pathologies of one of these pathways are clinically often accompanied by disruption of others [30]. Besides direct modulation of metabolic pathways, cardiovascular health in inflammatory diseases may be worsened also by oxidative stress. The excessive presence of reactive oxygen species (ROS) oxidates biomolecules like lipids, proteins, DNA or membranes and alters their function [31,32].

Some exogenous factors, like diet, exercise habits, environment, or genetic factors can cause lipid dysregulation, hyperlipidaemia, and obesity, disrupt glucose metabolism, and cause insulin resistance, often accompanied by hypertension. All these characteristics are often seen together in so-called metabolic syndrome (MetS) that can lead to atherosclerosis and finally to serious CVD events [33]. The adipose tissue, present in obese patients in excessive amount, represents an important energy depot but acts also as an endocrine gland. By releasing hormones (leptin, cortisol), cytokines (tumour necrosis factor α (TNFα)), or substrates (FFAs), adipocytes may influence a variety of tissues, alter glucose homeostasis and cause insulin resistance, which may progress to type 2 DM (T2DM) and its concomitant effects. Decreased responsiveness of adipose tissue to insulin stimulation in patients with insulin resistance may enhance FFAs release into the circulation, thus accelerating lipotoxicity, activating pro-inflammatory cytokine cascades, and promoting oxidative stress [34]. This may also partially explain the higher incidence of NAFLD in diabetic patients [35,36,37]. Despite some studies not considering NAFDL as a risk factor for T2DM development [38], others suggest a bidirectional relationship, meaning both pathological conditions (T2DM and NAFLD) accelerate the progression of each other [39]. Besides that, in T2DM, insufficient suppression of glucose output from the liver and decreased glucose transporter type 4 (GLUT4)-mediated glucose transport and metabolism in skeletal muscle or adipocytes, caused by insulin insensitivity of tissues, are typical [34]. Consequently, increased levels of plasma glucose associated with many unwanted effects like excessive protein glycation are observed. The abnormal lipid profile of diabetic patients includes decreased HDLc, increased amount of other lipoprotein particles and enhanced lipid peroxidation [40]. Altogether, pathologic conditions presented in T2DM lead to endothelial dysfunction, atherosclerosis progression, and increased CVD risk [3].

The term NAFLD denotes a spectrum of chronic liver diseases with potential progression from simple steatosis, characterised by TAG accumulation, through inflammation in NASH, to irreversible liver damage in fibrosis and cirrhosis. Several NAFLD-related factors are considered to contribute to CVD risk including dysregulation of lipoprotein metabolism leading to atherogenic dyslipidaemia [41], as low HDLc and higher apoB/apoA-I ratio [42,43]; altered glucose metabolism [44,45]; altered gut microbiome [46]; and chronic inflammation [47]. Meta-analysis revealed that early stages of NAFLD are associated with an increased risk of major adverse CVD events, but only the presence of inflammation in NASH, a more severe stage of NAFLD, is related to CVD mortality [1]. One of the factors involved in the pathogenesis of NAFLD is the composition of gut microbiota. Unfavourable alterations of gut microbiota, called dysbiosis, influence NAFLD, inflammatory bowel disease (IBD), type 2 diabetes mellitus (T2DM), and atherosclerosis in a negative way [6,48,49,50]. Impaired mucosal barrier function resulting in increased permeability in dysbiosis allows enhanced release of microbe- and pathogen-associated molecular patterns (MAMPs and PAMPs) into the circulation, activating Toll-like receptor (TLR) signalling. TLR activation leads to decreased cholesterol efflux and to overproduction of pro-inflammatory cytokines, such as TNFα, IL-1β, inducing systemic inflammatory response [51].

Similarly, as in previous diseases, the microbial imbalance can play a crucial role also in the pathogenesis of PD, the chronic bacterial inflammatory disease of the oral mucosa. Blood-borne pathogens and gut dysbiosis mediated by swallowed oral bacteria in PD [5,7], can elevate the inflammatory burden. Increased prevalence of CVD, T2DM, IBD, NAFLD, and RA was observed in PD patients [7,52]. In addition to systemic inflammation, triggering an antibody response against citrullinated proteins by *Porphyromonas gingivalis* in PD argues for the causal relationship between RA and PD [7].

RA is an autoimmune disease characterised by systemic low-grade inflammation and symmetric polyarthritis, proceeding to joint deformability and extra-articular damage, especially in the brain, liver, or lungs. Despite the lower levels of lipoproteins, patients with active RA have markedly increased CVD risk, in contrast to the general population [2]. This so-called “lipid paradox”, observed years before disease onset is supposed to be mainly the result of increased catabolism of lipoproteins, and related to qualitative alteration in lipoproteins, especially HDL [53]. Pro-inflammatory cytokines IL-1β, IL-6, and TNFα, released into systemic circulation during active RA, are considered to be the main contributors to atherosclerosis due to their impact on lipid and lipoprotein metabolism, and the biology of the artery wall [54]. Interestingly, the increase in lipoproteins, including total cholesterol (TC) and LDL, following the effective treatment of RA, is the marker of disease control associated with lower inflammatory status, and better expectations from the cardiovascular point of view [2,8,53].

Not only in RA but also in other rheumatoid diseases such as SLE, significantly increased carotid intima-media thickness (IMT) in comparison to the general population was observed, indicating subclinical atherosclerosis [48]. SLE, similarly to RA, is a chronic autoimmune inflammatory disease with unclear aetiology. The presence of autoantibodies against cell nucleus components, defective clearance of apoptotic cells and immune complexes, dysregulated and hyperreactive immune system or chronic inflammation are typical features seen in SLE patients [55]. In plasma of SLE patients, increased concentrations of IL-6 and monocyte chemoattractant protein-1 (MCP-1), cytokines associated with adverse lipid profile and atherosclerosis, were detected [56]. Antibodies against lipoprotein lipase (LPL), elevated TC, LDL, TAGs, and lower HDL levels are other factors contributing to atherosclerosis and organ damage in SLE patients [57]. Elevated CVD risk and atherosclerosis susceptibility in SLE patients were confirmed in clinical studies [58,59].

Psoriasis is an autoimmune disease with skin manifestation. Chronic inflammation leads to epidermal hyperplasia and the formation of typical skin lesions. In some patients, psoriasis may be accompanied by joint inflammation similar to that seen in RA, and such a pathologic condition is called psoriatic arthritis. Despite some differences in genetic background and clinical features of these diagnoses, RA, psoriasis but also psoriatic arthritis share abnormal lipid profiles and similar co-morbidities like NAFLD, infections, depression, anxiety, or cardiovascular complications [60]. Five years before the onset of psoriasis, a significant decrease in TC, LDL, and HDL, but higher mean TAG levels compared to non-psoriatic controls were observed [61]. Another study revealed elevated TAGs, cholesterol, and LDL but lower HDL in the serum of psoriatic patients when compared to healthy controls [62]. A significant increase in the incidence of cardiovascular complications was observed in patients with psoriatic arthritis or in psoriatic patients [63,64].

CKD is defined as abnormalities of kidney structure or functions lasting longer than 3 months. Estimation of glomerular filtration rate and albuminuria, the main diagnostic markers of CKD, shows increased CVD risk in patients with lower estimated glomerular filtration rate (eGFR) and higher albuminuria. It may seem that increased CVD risk in CKD patients may be only due to frequent underlying hypertension or DM in these patients. However, the meta-analyses clearly showed that impairment of kidney function is a predictive factor of CVD independent of these diagnoses [4,65]. The cardiovascular system in CKD may be disrupted by many mechanisms, like dyslipidaemia, oxidative stress, low-grade inflammation, increased vascular stiffness, endothelial dysfunction, electrolyte abnormalities, and others [66]. Dyslipidaemia in CKD patients is not characterised by particularly pro-atherogenic quantitative changes, but rather by qualitative modifications of components of lipid metabolism, like dysfunctional HDL and oxidised LDL (oxLDL) particles, modification of lipid content or particle size of particles, or some posttranslational modifications like glycation or oxidation [67,68].

To sum up, inflammatory conditions and altered lipid and glucose metabolism may influence the development of atherosclerosis. The progression of atherosclerosis was shown to be closely related to the progression of chronic inflammatory diseases. Moreover, as seen in numerous studies, a reciprocal relationship exists between various inflammatory diseases. Therefore, we can conclude that any evocation of a systemic inflammation influences many systems where they can cause or exacerbate inflammatory pathologies [1,5,6,7].

## 3. The Pathophysiology of Atherosclerosis

Etiopathogenesis of atherosclerosis, the dominant cause of CVD, is multifactorial. It includes oxidative stress, endothelial dysfunction, abnormal lipid metabolism, aggregation of thrombocytes, dysregulation of vascular tone, inflammation, and proliferation of vessel cells, which was discussed in detail elsewhere [69,70,71,72]. Shortly, dysfunction of ECs and the retention of apoB-containing lipoproteins in the subendothelial space are initial and crucial steps in the pathogenesis of atherosclerosis. Endothelium, the inner monolayer of the blood vessels, fulfils a broad variety of functions: (i) creates a barrier between circulating blood in vessels and the rest of the vessel walls, (ii) plays a role in vascular homeostasis, (iii) regulates vascular permeability and tone, (iv) regulates cell proliferation and (v) inflammation, (vi) produces chemical mediators influencing other cells such as monocytes, platelets, and vascular smooth muscle cells (VSMCs) [69]. Risk factors associated with increased oxidative stress and inflammation such as DM, dyslipidaemia, hypercholesterolemia, RA, SLE alter the balance in ECs by multiple mechanisms. Increased ROS induce cell damage by activation of pro-inflammatory transcription factor nuclear factor κ-light-chain enhancer of activated B cells (NF-κB), peroxidation of membrane lipids, and decreasing the bioavailability of nitric oxide (NO) [72]. Increased transcription of NF-κB-regulated genes, e.g., cytokines and adhesive molecules in dysbalanced ECs (vascular cell adhesion molecule 1 (VCAM-1), intercellular adhesion molecule 1 (ICAM-1), and selectins) results in chemotaxis and adhesion of circulating monocytes. The expression of other NF-κB-dependent genes, MCP-1 and IL-8, by ECs and monocytes, enhances this process by stimulating of monocyte migration into the subendothelial space. Other factors produced by ECs, such as macrophage-colony stimulating factor (M-CSF) and granulocyte macrophage-colony stimulating factor (GM-CSF), promote the differentiation of monocytes to macrophages, the main cell population in atherosclerotic plaques. Oxidative modification of LDL by ROS, myeloperoxidases (MPOs), and lipoxygenases (LOXs) results in the formation of oxLDL, inducing local inflammation [73]. In macrophages, continual intake of lipoproteins, especially those altered by oxidation or glycation, mediated by several receptors including lectin-like oxLDL Receptor-1 (LOX-1), scavenger receptor A (SR-A), scavenger receptor B1 (SR-B1), TLR4, cluster of differentiation 36 (CD36) and CD38 [69] leads to their transformation into foam cells [70,71,74]. These cells produce IL-1β stimulating VSMCs to produce also pro-inflammatory IL-6, which in turn regulates the expression of CRP (acute-phase protein, marker of systemic inflammation) in the liver. Thus, the triggering of macrophage inflammatory pathways boosts the progression of pathological events in atherosclerosis *via* the increase in oxidative stress, oxidation of LDL, EC activation, cytokine/chemokine secretion, and monocyte recruitment. Additionally, by infiltration of VSMCs into the intima and proliferation due to macrophage-derived chemoattractants, fatty streaks progress to a fibrous fatty lesion, making regression of atherosclerosis less likely [69]. In atherosclerotic lesions, endothelial injury, and dysfunction lead to a higher rate of apoptosis of ECs, contributing to the progression and pathophysiology of CAD [75]. With the increase in the cell volume of the intima, vascular remodelling results in the creation of stable fibrous plaque. The inflammatory environment leads to the formation of the vulnerable plaque, whose rupture causes thrombus formation accounting for the majority of CVD events. Elevated levels of CRP, IL-1β, IL-6 and TNF-α in patients with chronic inflammatory diseases stimulate the initiation and accelerate the progression of atherosclerosis, increasing markedly risk of CVD events [72,76]. Inflammation is considered the main contributor to atherogenesis by its impact on lipoprotein metabolism and the biology of the vessel wall.

## 4. Cardioprotective Effect of HDL Particles

A cardioprotective effect of HDL particles is given by influencing the pathogenesis of atherosclerosis through numerous beneficial mechanisms comprehensively reviewed elsewhere [69,77,78,79].

RCT is the most contributing and the best characterised beneficial molecular mechanism of HDL in terms of atherosclerotic progress. RCT represents a process of cholesterol sequestering from extrahepatic tissues, including macrophages localised in atherosclerotic plaques, into the liver. RCT starts with the transfer of free non-esterified cholesterol from the cells to HDL particles. This cellular cholesterol and phospholipid efflux is enabled by the interaction of ATP-binging cassette transporter A1 (ABCA1) in the cell membrane with lipid-free apoA-I or pre-β-HDL particles. Other receptors, ABCG1 and SR-B1, are involved in the cholesterol efflux to mature HDL. SR-B1, a receptor able to mediate bidirectional lipid/cholesterol flux, recognises an amphipathic helix present in the structure of all HDL-associated apolipoproteins, but only under lipidated status [80]. Cholesterol can also move from the plasma membrane of macrophages to HDL through passive diffusion [69]. Free cholesterol is esterified by lecithin-cholesterol acyltransferase (LCAT) and placed into the centre of the HDL particle. The next step, hepatic cholesterol delivery, can be accomplished by two routes: direct or indirect transport. Direct cholesterol transport is mediated mainly by hepatic SR-B1. The indirect cholesterol transport, a more preferred way of cholesterol clearance, is facilitated by cholesteryl ester transfer protein (CETP). CETP is exchanging cholesterol in HDL for TAGs present in very-low-density lipoprotein (VLDL) or LDL. These apoB-containing lipoprotein particles are sequentially recaptured by the liver mostly *via* LDL receptors. Cholesterol can be then used for the synthesis of VLDL, or it is excreted as free cholesterol or bile acids [12,13,20].

Even though the most protective effects of higher levels of HDL are connected to RCT, HDL also exhibits many other beneficial functionalities, such as anti-inflammatory, antioxidant, anti-apoptotic, endothelial-protective, or anti-thrombotic. Although macrophage cholesterol efflux by HDL alone significantly reduces inflammation, additional anti-inflammatory functions of HDL are playing an important role in atheroprotection. HDL inhibits the conversion of macrophages to the inflammatory M1 phenotype and promotes the anti-inflammatory M2 phenotype, regulates the expression of adhesive molecules on leukocytes and ECs (VCAM-1, ICAM-1), and suppresses VSMC proliferation and VSMC-derived secretion of monocyte MCP-1 in atherosclerotic plaques. These effects are mediated by multiple mechanisms, such as influencing numerous signalling pathways, enzymes, and transcription factors [19,78,81]. Anti-inflammatory and antioxidant functions of HDL are put into action by different HDL-associated apolipoproteins (apoA-I, apoCI-IV, apoE, apoJ and others), and by enzymes with antioxidant activity, such as PON1, glutathione peroxidase (Gpx-3) and lipoprotein-associated phospholipase A2 (Lp-PLA2), or by microRNA (miRNA) [21,82,83]. HDL directly inhibits the oxidation of LDL or other particles containing phospholipids. Circulating HDL accumulates oxidated phospholipids from LDL and cells, such as hydroperoxides, lysophophatidylcholine and F2-isoprostane. Prevention of forming of oxidated lipids and lipoproteins is secured by the hydrolysis of oxidated phospholipids by enzymes PON1, Gpx-3 and Lp-PLA2. As a result, LDL oxidation and cell oxidative status are diminished [69,84], preventing in turn the generation of the pathological inflammatory process [15].

NO produced by endothelial NO synthase (eNOS) is an important factor regulating endothelial function with vasodilatory, antiplatelet, antioxidant, and anti-inflammatory properties [85]. HDL is able to activate eNOS synthesis in ECs through SR-B1 and for this interaction, apoA-I binding seems to be essential [86]. Directly or *via* eNOS, HDL may also attenuate the ligand-independent SR-B1 mediated apoptosis of ECs. Even though apoptosis (induced *via* SR-B1) is necessary for the rapid elimination of damaged ECs to prevent damage to neighbouring cells, inappropriate apoptosis of ECs injured by oxLDL or ROS during chronic inflammation contributes significantly to the progression of CAD. HDL may prevent this excessive apoptosis by direct binding to SR-B1, which promotes endothelial repair [87,88,89]. The antiapoptotic ability of HDL is partly mediated *via* apoJ by activating phosphoinositide 3-kinase (PI3K)/Akt [75]. Besides the anti-apoptotic activity of HDL in ECs, by promoting efflux of cytotoxic oxysterols from macrophages, HDL may prevent their apoptosis, too [90]. Additionally, HDL prevents thrombosis by inhibiting of coagulation factors or by minimizing their cholesterol content *via* SR-B1, thus preventing their aggregation [78,91].

## 5. HDL Protein Components

HDL lipoprotein particles are highly heterogeneous. They differ markedly in their size, shape, function, lipidome, and proteome composition. The 5 subpopulations of HDL; HDL2b, HDL2a, HDL3a, HDL3b, and HDL3c; can be separated based on density and size using ultracentrifugation and non-denaturing polyacrylamide gradient gel electrophoresis [21,92].

The basic structure of HDL consists of a surface monolayer of polar lipids (phospholipids, non-esterified cholesterol) solubilised with the help of apolipoproteins, and a central hydrophobic core containing nonpolar lipids (TAGs and cholesteryl esters (CEs)). HDL also contains many other protein components. Interestingly, the number of protein components of HDL is much higher in comparison to other lipoprotein particles [92]. According to Davidson and Shah, groups from the University of Cincinnati, more than 200 protein components associated with HDL were identified in at least 3 proteomic studies, while LDL likely contains only 22 [93,94,95]. The concrete count of HDL-associated proteins differs from study to study, there is also great inter-individual variability of HDL proteome in humans [96]. HDL proteome changes according to actual physiological or pathological conditions in an organism, and also the contamination could distort the results of analysis, hence specifying the exact HDL proteome is difficult. In general, major part of HDL proteins represent apolipoproteins for example apoA-I, apoA-II, apoA-IV, apoA-V, apoC-I, apoC-II, apoC-III, apoC-IV, apoD, apoE, apoF, apoH, apoJ (clusterin), apoL or apoM, which regulate lipid metabolism and some of them participate also in acute inflammatory response or complement regulation. Besides that, HDL may contain a variety of enzymes with different functionality (PON1 and PON3, LCAT, Lp-PLA2, or Gpx-3), lipid transfer proteins (phospholipid transfer protein (PLTP) or CETP), proteinase inhibitors (alpha-1-antitrypsin (AAT) and haptoglobin (Hp)-related protein), acute phase proteins (SAA, ceruloplasmin, fibrinogen, hemopexin (Hx), transferrin, complement components) and many others [92,97].

## 6. Protein Components of HDL Relevant to Atherosclerosis

### 6.1. apoA-I

ApoA-I, the most abundant protein in HDL, represents approximately 70% of total HDL protein content. It is mainly synthesised in the liver and small intestine. Nearly all HDL lipoproteins contain apoA-I. However, apoA-I is not unique to HDL, as a small amount of this protein was detected also in VLDL and chylomicrons. The scope of functions of apoA-I is quite comprehensive. The highly flexible structure of apoA-I, enabling a dynamic shift between lipid-bound and lipid-free states, is rendering the high effectivity of apoA-I in the interaction with ABCA1 and ABCG1 receptors on membranes and removal of phospholipids and cholesterol from foam cells [98,99]. ApoA-I is necessary for many steps of RCT. It maintains the formation and stabilisation of HDL particles’ structure, mediates cholesterol efflux by interaction with ABCA1, activates LCAT, and binds as a ligand to SR-B1, promoting hepatic clearance of peripheral cholesterol [24,99]. Besides that, apoA-I showed anti-inflammatory and endothelial-protective properties *in vivo*, for example by decreased ICAM-1 and VCAM-1 expression, and NF-κB signalling in ECs [100,101], attenuation of neutrophil activation [102] or participation in SR-B1 mediated eNOS stimulation [103]. Overexpression of apoA-I diminished systemic inflammation and its consequences induced by lipopolysaccharide (LPS) in mice model, in which apoA-I administration reduced liver CD14 expression and serum levels of inflammatory cytokines (TNFα, IL-6, IL-1β), and lessened organ damage [104]. Low apoA-I levels during childhood seem to be a predictive marker for later artery IMT development in adulthood [105]. ApoA-I levels are typically reduced in HDL of patients with chronic inflammatory diseases [106]. Moreover, apoA-I protein structure and function are affected under chronic inflammatory conditions. Additionally, apoA-I mutations, such as K107del, L144R, A164S and L178P, affect the conformation and thermodynamic stability of the protein, impairing HDL function. The association of specific mutation of apoA-I (K107del) with increased atherosclerosis susceptibility underlines the role of apoA-I in atherosclerosis development [98]. Interestingly, L144R point mutation of apoA-I in heterozygous carriers does not increase CVD risk despite significantly reduced HDLc, inhibited LCAT activity, and SR-B1-driven cholesterol efflux [107,108]. This mutation has preserved ABCA1-mediated cholesterol efflux capacity (CEC) and endothelial-protective effect, in contrast to A164S apoA-I mutation, displaying decreased stimulation of EC migration probably due to reducing ABCG1-mediated CEC and activating endothelial receptors LOX-1, leading to blocked Akt activation (higher malondialdehyde (MDA), ROS) [107,108]. Nevertheless, apoA-I[A164S] heterozygous carriers have unaffected LCAT activation and normal lipoprotein levels, including HDL, but are at higher risk of CVD-linked mortality [108]. These results indicate that the preservation of endothelial monolayer integrity and function (vascular protective role of HDL) is more important in atheroprotection than the quantity of HDL lipoprotein [89].

### 6.2. apoA-II

ApoA-II is the second major HDL protein constituent (approx. 15–20% of HDL proteins), but it was shown to be present in only about half of HDL lipoprotein particles. The main site for apoA-II synthesis is, as for apoA-I, the liver, and small intestine. Unlike apoA-I, the biological and physiological functions of apoA-II are unclear. ApoA-II is more hydrophobic than apoA-I. By interacting with apoA-I and other apolipoproteins, apoA-II is closely associated with the modulation of HDL metabolism and alteration of HDL size and conformation [109,110]. ApoA-II is proposed to play a role in the regulation of each step in HDL metabolism, even though the effect seems not to be strong. The results of studies that evaluate the effect of apoA-II on atherogenesis (in humans, in transgenic animal models, or in different *in vitro* models) are controversial [111,112,113]. By influencing the activity of enzymes and receptors active in removing lipid from the circulation, apoA-II was shown to alter the intermediate HDL metabolism, affecting the atherogenicity of lipid metabolism by both, detrimental (inhibition of LCAT and SR-B1) and beneficial effects (activation of hepatic lipase (HL) and inhibition of CETP) [111,112]. Enrichment of HDL with apoE, apoA-IV and apoA-V and other proteins is affected by the presence of apoA-II [112,114,115]. Castellani et al. (1997) [114] claimed overexpression of apoA-II in transgenic mice converts HDL to piHDL particles with decreased PON1 activity. Another study found a correlation between mesenteric fat thickness, an independent determinant of MetS, and increased carotid intima-mediate thickness, with apoA-II levels [116].

### 6.3. apoA-IV

ApoA-IV is the third most abundant HDL apolipoprotein, but it is not bound exclusively to HDL. It can be associated with other lipoproteins, such as chylomicron remnants or even circulate in a lipid-free form. ApoA-IV is the most hydrophobic apolipoprotein synthesized in the small intestine [92]. ApoA-IV provides a wide spectrum of physiologic functions [117]. At first, apoA-IV seems to have an important role in lipid absorption. Despite the apparent connection seen in cell culture experiments between apoA-IV expression and intestinal TAGs absorption and packaging, many *in vivo* studies using the traditional quantification methods showed no difference in absorption by apoA-IV knockout or the transgenic overexpression [118]. The explanation for this controversy could be the regional heterogeneity of intestinal absorption so that the apoA-IV effect could be observed only in specific regions of the small intestine [119]. ApoA-IV interferes with lipid, but also with glucose metabolism. In rats, knockout of apoA-IV led to increased glycolysis, decreased gluconeogenesis, and stimulated *de novo* lipogenesis, suggesting it may represent some link between these metabolic pathways [120]. Allergic patients have lower apoA-IV plasmatic levels. *In vitro* apoA-IV reduced eosinophil responsiveness and *in vivo*, it reduced hyperresponsiveness and airway eosinophilia in a mice model of house dust-induced asthma and inflammation in dextran sulphate sodium-induced colitis [121]. ApoA-IV alleviated the LPS-induced inflammation in macrophages [122]. These facts suggest that apoA-IV probably acts also as an endogenous anti-inflammatory molecule. ApoA-IV prevents thrombosis by inhibiting platelet aggregation and hyperactivity [123], enhances cholesterol efflux by ABCA1 [124], activates LCAT [125], prevents the oxidation of LDL molecules [126], and attenuates the risk for the development of atherosclerosis [127]. The ability of apoA-IV to inhibit lipid peroxidation seems to depend on its isoform [126]. Increased glycosylation of apoA-IV in patients with DM was associated with increased CVD risk [128].

### 6.4. apoE

ApoE is the fourth most abundant HDL apolipoprotein. Most of the plasma apoE is liver-derived. ApoE is also synthesised in other multiple tissues, such as macrophages, adipocytes, and astrocytes in the central nervous system (CNS), with unique functional attributes and mostly local effects. ApoE is a multifunctional protein associated in plasma with almost all lipoprotein particles, regulating many steps in lipid and lipoprotein homeostasis [129], besides its pleiotropic role including cell proliferation, inflammation, oxidative stress, macrophage and neuronal cell homeostasis, and others [129,130]. ApoE exerts local anti-inflammatory effects by promoting the conversion of macrophages from the pro-inflammatory M1 to the anti-inflammatory M2 [131]. The C-terminal domain of apoE contains amphipathic α-helices important for interaction with lipids, which represent a high-affinity lipid binding region, typical for exchangeable apolipoproteins. The N-terminal part of apoE comprises the receptor-binding region, including LDL receptor (LDLR), LDLR-related protein (LRP), the heparan sulphate proteoglycans (HSPGs) and ABCA1 [132]. Three human *APOE* isoforms (designated apoE2, apoE3, and apoE4) have different structures and functions, such as different receptor affinities and lipoprotein-binding preferences or altered anti-inflammatory effectivity [133].

Subspecies of HDL containing apoE account only for 5–10% of total plasma HDL either with or without apoA-I. The closest structural apolipoprotein of apoE is apoA-I [134]. ApoE-HDL has anti-atherogenic, anti-inflammatory, and antioxidant properties. ApoE regulates the metabolism of HDL on multiple levels. Experiments on apoE^−/−^ × apoA-I^−/−^ mice revealed that apoE can promote *de novo* biogenesis of HDL in a manner independent of functional apoA-I. This process includes the participation of ABCA1 and LCAT [135,136,137]. Macrophage-derived apoE intervenes into the kinetic of RCT [138,139]. Interaction of apoE with ABCA1 and SR-B1 on macrophages mediates cellular cholesterol efflux [132,140]. ApoE accommodates the size expansion of HDL due to internalisation of CEs into the HDL’s core in conjunction with the action of LCAT [136,138]. While apoA-I binds to the fatty acid acyl chains of phospholipids, apoE interacts with polar phospholipid head groups. As a result, the size and CE content of the HDL can be significantly increased in the presence of apoE in comparison to only apoA-I containing HDL [138]. VLDL and intermediate-density lipoproteins (IDL) particles enriched with apoE, a high-affinity ligand for several hepatic lipoprotein receptors, are cleared from the circulation much more quickly than those without apoE in humans. Similarly, apoE mediates HDL holoparticle clearance by the liver [141]. ApoE accelerates the regeneration of small HDL particles by selective removal of CEs not only by SR-B1 but also by other hepatic cell-surface receptors [136,140]. The selective uptake pathway is impaired in the liver and adrenal glands in apoE^−/−^ mice [142]. In a large prospective population-based study, the apoE concentration in HDL is inversely related to the risk of CHD but only in the absence of apoC-III [141].

ApoE positively influences the activity of enzymes involved in lipoprotein metabolism, such as LCAT, HL, and CETP [143]. The binding of PON1 to reconstituted apoE-HDL stabilised the enzyme and stimulated the antioxidant potential of PON1, measured by inhibition of LDL oxidation [144]. ApoE-HDL was found to have a relatively high Lp-PLA2 activity. This activity was lower and negatively correlated with MDA levels in patients with polycystic ovary syndrome [145]. ApoE-HDL particles were identified as the major HDL subclass able to inhibit agonist-induced platelet aggregation [146]. The proposed mechanism of apoE-driven anti-platelet actions is *via* L-arginine: NO signal transduction pathway, by enhancing the production of NO [147]. Moreover, apoE can mediate the antimitogenic effect of HDL on VSMCs by inducing cyclooxygenase 2 (COX2)-dependent synthesis of prostacyclin and miR-145-dependent LOX mRNA inhibition. As a result, VSMCs proliferation and expression of extracellular matrix genes, which play a role in vascular remodelling and atherosclerosis, are inhibited [148,149]. ApoE polymorphism, especially ε4 allele (E4/E4 or E4/E3 phenotype), is connected to susceptibility to Alzheimer’s disease and atherosclerosis [130]. ApoE^−/−^ mice develop spontaneously atherosclerosis [77], highlighting the significance of this apolipoprotein in atheroprotection.

Other proteins associated with HDL are in minor abundance.

### 6.5. apoA-V

ApoA-V is a protein synthesised almost exclusively in the liver, which is, despite its relatively low concentrations, probably the key regulator of serum TAG concentration. In circulation, it is distributed in HDL, VLDL, and chylomicrons, but not in LDL [150]. In patients with apoA-V deficiency, hypertriglyceridaemia and low plasma HDL were observed [151]. Hypertriglyceridaemia was observed also in apoA-V deficient mice, in which administration of apoA-V- containing HDL particles significantly reduced TAG levels [152]. ApoA-V-enrichment of HDL in apoC-III transgenic mice was associated with lower apoC-III and higher apoA-I and apoE content in HDL, increased LCAT activity and increased cholesterol efflux, suggesting the role of apoA-V also in modulating HDL maturation and cholesterol metabolism [153]. The effect of apoA-V on HDL metabolism along with its TAG lowering activity may have a protective effect on atherosclerosis development [154]. Even though the results from the mice-model experiment classified apoA-V as positive inflammatory acute-phase reactants [155], other animal or human studies observed reduced apoA-V level during inflammation [156,157].

### 6.6. apoC

ApoC (apoC-I, apoC-II, apoC-III, apoC-IV) are small apolipoproteins, predominantly synthesised in the liver and distributed in circulation between lipoproteins, like chylomicrons, VLDL and HDL [158,159,160]. In normolipidemic subjects, apoC-II and apoC-III are usually equally distributed between VLDL and HDL, whilst apoC-I is bound mainly to HDL and apoC-IV to VLDL [161]. ApoC-II and apoC-III (together with apoE) are readily transferred between VLDL and HDL dependending on TAG metabolism. In hypertriglyceridemic subjects, apoC-II was found to be predominantly bound to apoB-containing lipoproteins (VLDL, LDL) [158,162]. However, even in healthy subjects, there can be marked individual differences in this distribution [158]. ApoCs play an important role in lipid and TAG metabolism. ApoC-I influences lipid metabolism *via* CETP inhibition [163], reduction in FFAs release from lipoproteins mediated by LPL [164] and their cellular uptake [165]. Besides that, *via* enhancement of immune response, apoC-I protects the body against infections [166]. ApoC-III attenuates the rate of the size increase and clearance of apoE-containing HDL [141]. ApoC-III inhibits the binding of apoE to various receptors, resulting in delayed clearing of HDL, as it is also known for apoE containing VLDL and IDL [141,167]. Similarly, as apoC-I, it also promotes inflammation by alternative inflammasome activation [168]. Unlike LPL-inhibitors apoC-I and apoC-III, apoC-II is able to activate LPL, thus increasing the release of FFAs from lipoproteins, and allowing their uptake by tissues [169]. However, excess amount or deficiency of apoC-II lower LPL activity and affect HDL maturation which may manifest in altered HDL subclass distribution [170,171]. ApoC proteins interfere also with LCAT activity, where apoC-I seems to be an activator, whilst apoC-II and apoC-III inhibit LCAT [172]. The nine-year follow-up study found an association between CVD mortality and low apoC-II and apoC-III levels in T2DM patients [173]. On the contrary, a case–control study marked elevated serum apo C-II as a potential risk factor for CHD [174]. However, these reports do not distinguish between apoCs on HDL vs. VLDL. HDL containing apoC-III was, according to some studies, associated with a higher risk of CHD, whereas HDL not containing apoC-III was associated with lower CHD risk [115,141]. More epidemiologic data are required to specify a causality between apoC-II and CVD. The newest member of apoC family is apoC-IV. Compared to other family members, apoC-IV represents only a minor component of plasmatic lipoproteins [161]. ApoC-IV overexpression disrupted lipid metabolism in Huh-7 cell line, suggesting its role in hepatic steatosis [175]. The *APOC4* polymorphism seems to be associated with CVD risk [176].

### 6.7. apoJ

ApoJ, known also as clusterin, is present in a small subpopulation of HDL together with apoA-I and PON1. Most of apoJ circulates in plasma unbound and only 20–30% of apoJ is associated with lipoproteins [177]. The structure of apoJ enables the binding of specific cell-surface receptors and lipids. ApoJ influences numerous physiological and pathological processes, including transport of lipids, cell differentiation, apoptosis, cell adhesion, regulation of the immune system, and oxidative stress [178,179]. This heterodimeric glycoprotein belongs to positive acute phase proteins, as its mRNA and protein expression is induced by endotoxin, TNFα or IL-1 [180,181]. Fasting plasma apoJ levels correlate with the parameters of adiposity and CRP levels in healthy adults [182]. As a sensitive sensor of oxidative stress, apoJ works as a chaperone, favouring correct protein conformation or binding to misfolded proteins for their clearance. Using the chaperone activity, apoJ can stabilise HDL proteins (such as apoA-I, PON1, LCAT, Lp-PLA2). It can also bind and sequester oxidised lipids, further contributing to the preservation of HDL’s anti-inflammatory, antioxidant and antiapoptotic properties. Other documented actions of apoJ are inhibition of monocyte migration, proliferation of VSMCs, or complement activation [178]. Moreover, reconstituted recombinant HDL-recombinant apoJ nanoparticles mediate cholesterol efflux in a dose-dependent manner from cultured mouse macrophages (the same percentage as apoA-I *in vitro*) [183]. Interestingly, apoJ binds to subpopulations of LDL with increased negative charge LDL(−), as small dense LDL, which are increased in patients with T2DM and hypertriglyceridaemia, to prevent atherogenic modifications of LDL [177]. *APOJ* single nucleotide polymorphism (1598delT) was found to be associated with abnormal levels of HDL and risk factors for CAD [184].

Plasma levels of apoJ correlate with adiposity, T2D, ageing, developing CAD, oxidative stress, and systemic inflammation [181,182,185]. The gene expression is dramatically increased in injured tissues, atherosclerotic lesions, or in the loci of neurodegeneration in Alzheimer’s disease (AD) [181,186,187]. Together with apoE is apoJ most abundant apolipoprotein in the brain, both associated with amyloid beta. The most widely studied function of apoJ is its positive role in AD pathology [183,186]. In the development of atherosclerosis, the role of apoJ remains controversial, which can be explained by the opposite effect of two alternative splice forms, secretory apoJ and nuclear apoJ [177]. Colocalization of apoJ with apoA-I and PON1 in aortic lesions was observed, indicating that a part of present apoJ can be attributed to HDL particles [187]. Serum levels of apoJ were proposed as markers of vascular damage. Immunolocalization of apoJ increases with the atherosclerotic lesion progression and is positively correlated with serum TNFα concentration or smoking, which is thought to cause oxidative damage [181,187]. In numerous atherosclerosis-prone mice models (C57BL/6J, apoE^−/−^, LDLR^−/−^) or in rabbits under inflammatory conditions or an atherogenic diet, a marked decrease in PON1 activity accompanied by a dramatic increase in apoJ was observed. Additionally, in normolipidemic patients with CAD (without DM) despite normal HDL levels, the apoJ/PON1 ratio was significantly increased in comparison to healthy controls. The HDL from these patients lost its ability to protect against LDL oxidation *in vitro* [178]. It was proposed that the explanation of increased apoJ in HDL is to prevent damage of HDL’s subunits by its chaperone activity and sequestering lipid hydroperoxides, and to preserve HDL’s anti-inflammatory, antioxidant and antiapoptotic properties.

### 6.8. apoM, apoD and apoF

ApoM is another HDL-associated apolipoprotein. Studies suggest that enrichment of HDL particles with apoM increases HDL capacity to prevent LDL oxidation, stimulates cholesterol efflux from macrophages, decelerates atherosclerotic lesions-formation in LDL deficient mice [188,189], and improves the anti-inflammatory effect of HDL [190]. ApoM is also an essential transporter of important signal molecule for immune and vascular system, sphingosine-1-Phosphate (S1P). Experiments in apoM-transgenic mice revealed its ability to disrupt TAG turnover [191]. Lower apoM correlates with increased endothelial dysfunction and disease severity in SLE patients [192].

Other apolipoproteins presented in HDL could also have an important role in lipid metabolism and HDL function. For example, apoD seems to influence the LCAT activity [193] or apoF probably modulates RCT and the function of CETP [194].

### 6.9. CETP

CETP is a protein synthesised in the liver and adipose tissue. It circulates in plasma bound mainly to HDL. CETP is activated by apolipoproteins (apoA-I, apoA-II, apoA-IV, apoE, apoC-III) [195]. It catalyses the exchange of CEs from HDL to apoB-containing particles (VLDL, LDL) in exchange for TAGs. This reaction is an essential part of RCT by which the CEs exchanged from HDL could be cleared from circulation in apoB lipoproteins. At the same time, a reduction in the number of HDL particles occurs, which could be considered pro-atherogenic [196]. Despite the clinical observation of increased HDL levels in patients with CETP deficiency [197], the conclusions from human studies about the relationship between CETP deficiency and CVD risk are not uniform, suggesting its probable complexity. Some studies indicate improved [198] or unchanged [199], and other worsened [200,201] CVD risk or its predictive parameters in patients with CETP deficiency. Besides the effect on lipid metabolism, enhanced CETP activity seems to be associated also with increased risk for venous thromboembolism *via* direct binding to factor Xa [202].

### 6.10. Lp-PLA2

Lp-PLA2 also known as platelet-activating factor-acetyl hydrolase (PAF-AH), is an enzyme with phospholipase A2 activity. Lp-PLA2 circulates primarily with LDL and to a lesser extent with HDL in plasma. Plasma isoform is synthesized mainly in macrophages, whereas several soft tissues express specific liver isoform [203]. The synthesis is associated with the level of inflammation (estimated by the concentration of CRP). The exact role of Lp-PLA2 in atherosclerosis pathogenesis is not well understood. This enzyme prevents LDL oxidation by hydrolysing platelet-activating factor (PAF) (an ether phospholipid) and PAF-like oxidised phospholipids, potent pro-inflammatory lipid mediators involved in atherosclerosis. HDL-associated Lp-PLA2 was reported to protect LDL from oxidation and inhibit cell stimulation by oxidised LDL, thus exhibiting anti-inflammatory and anti-atherogenic functions [204]. On the contrary, some evidence for pro-atherogenic and pro-inflammatory role of Lp-PLA2 was observed. Lp-PLA2 generates bioactive compounds like lysophosphatidylcholine or oxidized non-esterified fatty acids with pro-inflammatory properties and in experimental settings, Lp-PLA2 inhibition diminished these negative effects [205]. In addition, Lp-PLA2 mediated leukocyte activation and inflammatory responses *in vitro* [206]. These conflicting observations led to the hypothesis that the protective vs. pro-inflammatory and pro-atherogenic character of Lp-PLA2 activity depends on the lipoprotein particle, to which Lp-PLA2 is currently bound to. Studies suggest anti-atherogenic activity is tied to HDL-bound Lp-PLA2 [207]. In population studies, increased Lp-PLA2 mass and activity correlate with increased CVD risk [208]. Some studies use the quantification of plasma activity of Lp-PLA2 instead of isolation of HDL and quantification only of the HDL-associated activity. Thus, measuring only the plasmatic activity of this enzyme could be confusing and may not match the real situation in HDL.

### 6.11. PLTP

PLTP is a protein with a structure highly similar to CETP. It is expressed in a variety of tissues, mainly in the liver and intestine, and similarly to CETP, its expression is upregulated *via* liver X receptors (LXRs) when the cholesterol content rises. Interaction of PLTP with apoA-I, apoA-II and apoE affects its binding to HDL and its activity (active vs. inactive form) [209,210]. PLTP provides the transfer of phospholipids and other molecules (α-tocopherol) between lipoproteins, contributing to HDL maturation [211] and promoting ABCA1-mediated efflux of cholesterol and phospholipids from cells, which in general improves RCT [212,213,214]. HDLc was reported to rise with increased PLTP activity. Nevertheless, increased systemic PLTP expression in mice has a pro-atherogenic effect [215,216,217], and an increase in PLTP also seems to act pro-coagulative [218]. The effect of PLTP on inflammation is not fully understood. In rheumatoid patients, PLTP has a direct pro-inflammatory effect on fibroblast-like synoviocytes [219] and its pro-inflammatory properties were observed also in several animal models [220,221,222]. On the other hand, mainly in the studies using LPS-induced inflammation, the anti-inflammatory effect of PLTP was observed [223,224,225]. During bacterial infection, PLTP plays an important role in innate immunity, prevents the growth of Gram-negative bacteria, and lowers LPS-induced toxicity [226,227]. PLTP deficiency in mice resulted in improved anti-inflammatory and antioxidant properties of HDL particles [228]. Gender-specific correlation (in men only) between higher plasma PLTP or lower plasma CETP activities, respectively, and CVD risk was found [229].

### 6.12. LPS-Binding Protein (LBP)

Another protein belonging to the same protein family as CETP and PLTP is LBP. LBP is the liver-produced, LPS-binding acute phase protein enabling the recognition of LPS by host cells and eliciting the immune response. Besides that, LBP probably mediates LPS transfer to HDL particles and its subsequent neutralisation [230] and, together with PLTP, exchanges a LPS between lipoproteins [231]. LBP may serve also as a marker of increased risk for atherosclerosis [232,233,234].

### 6.13. LCAT

LCAT is a glycoprotein with phospholipase A2 and acyltransferase activity. This liver-derived enzyme circulates in plasma either bound to lipoproteins HDL and, to a lesser extent LDL. LCAT activity facilitates the flux of unesterified cholesterol from peripheral cells by creating a gradient, thereby playing an important role in RCT. LCAT catalyses the conversion of cholesterol to CE on the surface of lipoproteins using fatty acids from phosphatidylcholine, converting it to lysophosphatidylcholine. CEs are moved from the surface to the hydrophobic core, enabling the maturation of nascent HDL into spherical HDL. ApoA-I, besides other apolipoproteins (apoA-II, apoA-IV, apoE, apoC-I), is an important activator of LCAT, whereas this activation is more effective in small HDL particles [235]. Mutations of LCAT causing loss of LCAT activity are associated with very low HDLc (5–10%) and LDL in the plasma due to accelerated catabolism of lipoproteins [236]. Although the ability to form mature HDL and esterify free plasma cholesterol is affected, these subjects do not exhibit a marked increase in CVD risk, presumably due to increased lipid-poor pre-β HDL, which acts *via* ABCA1 in RCT and is considered to be atheroprotective [236,237,238]. In compliance with it, LCAT expression in mice was inversely associated with ABCA1-dependent cholesterol efflux from macrophages. On the other hand, too high LCAT overload gives rise to very large HDLs rich in CEs (apoE-rich HDL1), dysfunctional in CE delivery to the liver, which can lead to increased atherosclerosis. Even though studies in murine models did not lead to conclusive results about the impact of LCAT deficiency and overexpression on atherosclerosis [238,239], indications are ascribing a more likely atheroprotective role of LCAT in more humanized animal models [240].

### 6.14. Gpx-3

Gpx-3, also known as glutathione selenoperoxidase 3, is a kidney-produced antioxidant enzyme associated exclusively with the plasma HDL fraction [241,242]. It is the only extracellular member of the Gpx family and probably the most important extracellular antioxidant enzyme in mammals. Gpx family contains several isotypes of selenium-containing enzymes, which catalyse the reduction in lipid peroxides and hydrogen peroxide and thus protect the body from low-level oxidative stress [243]. Antioxidant activity of peroxisome proliferator-activated receptor γ (PPARγ)-agonists, therapeutic agents used in T2DM patients which prevent oxidative stress-induced insulin resistance in these patients, is probably mediated by Gpx-3 [244,245]. Gpx-3 deficiency also leads to increased vascular dysfunction and risk of platelet-dependent thrombosis, suggesting its role in platelet and endothelial function [246]. Low levels of HDL and Gpx-3 are markers for increased risk for CVD mortality [247].

### 6.15. PON

Three PON isoforms: PON1, PON2, and PON3, all components of HDL, are able to hydrolyse lipid peroxides in LDL. PON1, the most studied among them, is a glycoprotein composed of 354 amino acids, synthesised in the liver and sequentially secerned into blood, where it is connected to HDL *via* the interaction with apoA-I and phospholipids [9]. The presence of apoA-II influences this binding [115]. Studies reveal that higher PON1 activity is associated with lower CVD incidence [83]. PON1 as a part of HDL possesses direct antioxidant, anti-inflammatory, antiapoptotic and antithrombotic effects. The enzyme has aryl esterase, paraoxonase, and lactonase activities, being able to degrade oxidized phospholipids and hydrolyse lactones from lipoproteins and cells. PON1, the major HDL antioxidant component, preserves HDL and LDL functionality by preventing their oxidation [84,248]. Incubation of human coronary and carotid lesions with PON1 reduced the lipid peroxide content in the lesions, which suggests a potential for the anti-atherogenic activity of PON1 [249]. Additionally, PON1 stimulates cholesterol efflux and regulates ABCA1 expression in macrophages [250]. According to some data, PON1 may also hydrolyse the PAF, a mediator of platelet activity and inflammatory pathways [251], however, no correlation between platelet aggregability and PON1 activity in thienopyridine-treated patients was observed [252]. PON1 is an important determinant of HDL to stimulate endothelial production of NO *via* endothelial receptor SR-B1. Inhibition of PON1 in HDL damages the ability of HDL to produce endothelial NO. Moreover, damped levels of NO impede inhibitory effects of HDL on the activity of pro-inflammatory NF-κB, expression of VCAM-1, and monocyte adhesion [21,253].

### 6.16. Transferrin

Transferrin is ferric ions binding glycoprotein synthesized in the liver, which represents the most important body’s iron pool, providing its transfer and maintaining the iron homeostasis [254]. It may circulate in plasma freely [255] or as a part of HDL particles [97]. After absorption or release from the reticuloendothelial system, the ferric ions are subsequently bound to transferrin, which transports it to tissues, such as the bone marrow for haemoglobin and erythrocytes production. Moreover, this transfer probably plays a role in the innate immunity system, where it reduces pathogens’ access to iron ions, necessary for pathogens’ growth [254]. Transferrin belongs to negative acute-phase proteins, which means its serum levels usually decrease during acute inflammation, although low levels are observed also during chronic inflammation [256,257]. Sequestration of iron ions by transferrin also prevents the iron-mediated formation of ROS, thus transferrin serves also as an antioxidant. *In vitro* study suggests, that decreased transferrin levels and its glycation status are associated with increased pro-oxidant iron effects and may enhance lipid peroxidation [258].

The following proteins are subunits of piHDL.

### 6.17. SAA

SAA are highly conserved proteins, named after identification as precursors of amyloid disease deposits. SAA belong to apolipoprotein family, predominantly synthesized in the liver, but also macrophages can be involved. Four different SAA isoforms are known in human genome. SAA1/SAA2 are major acute phase response (APR) isoforms, both commonly referred to as SAA. SAA3 is considered a pseudogene and SAA4, also found as an apolipoprotein of HDL, is constitutively expressed [259]. After induction of APR with various pro-inflammatory stimuli, including TNF, IL-1β, IL-6 and IFNγ, the concentration of SAA in plasma rises rapidly, reflecting the increased production and secretion. SAA play a role in innate immunity *via* binding to Gram-negative bacteria and mediating inflammation and chemotaxis to different cell types. On the other hand, SAA are important in a negative feedback loop to extinguish inflammation to prevent lethality due to sepsis [259,260].

Patients with chronic inflammatory diseases, such as visceral obesity, T2DM, RA, psoriasis, and CKD have increased levels of SAA (170, 181, 190). Levels of APR SAA correlate with the severity of inflammatory diseases, such as atherosclerosis and RA [260]. An increased all-cause and CVD mortality in association with high SAA concentration were demonstrated [261]. Plasma SAA was suggested to be a better predictor of future CVD events than CRP [262,263]. A more direct association with atherosclerosis was shown, as SAA has been detected within atherosclerotic lesions in coronary and aortic arteries in humans and mouse models [259,264]. Deficiency or suppression of all three acute phase SAAs: SAA1, SAA2.1, and SAA3 was necessary to significantly reduce atherosclerosis in apoE^−/−^ mice [265].

SAA circulate in plasma mostly as HDL component (~95%), in particular with HDL3. A smaller fraction of SAA can be associated with apoB-containing lipoproteins, LDL and VLDL, or can exist in a lipid-free form. Under physiological conditions, equilibrium between HDL-associated SAA and other forms of SAA exists [259,262]. However, diversity in HDL particle size and SAA distribution was observed in HDL obtained from patients with inflammation. Depending on the severity of the inflammatory response, SAA can become the major apolipoprotein on HDL (up to 87% of the total protein content of HDL). The binding of SAA to HDL leads to significant alterations in HDL protein and lipid composition, and to shift from anti-inflammatory to dysfunctional and pro-inflammatory HDL. SAA, having a lipophilic surface, exhibit increased binding affinity to HDL in comparison to apoA-I with a hydrophobic core. Therefore, apoA-I is readily replaced from HDL by SAA under inflammatory conditions, as shown in reports using material from human and mouse sources [259,266]. An alternative way of SAA-enriched HDL (SAA-HDL) generation, directly through ABCA1 in the liver, in a similar way to the biogenesis of normal HDL by apoA-I and apoE, was observed in mice [24].

Loss of the main structural protein of anti-inflammatory HDL may result in impaired apoA-I-associated anti-atherogenic actions. The association between SAA and RCT, a key process counteracting cholesterol accumulation in atherosclerotic lesions in arteries, was intensively followed up. During APR, HDL-mediated cholesterol efflux from the periphery, as well as HDL-mediated cholesterol (free or esterified) delivery to the liver is considerably decreased. Numerous studies of SAA-HDL or lipid-free SAA impact on RCT in humans with chronic inflammatory diseases and mice brought conflicting results. Evidence was reported both for the participation of SAA in cholesterol efflux *via* ABCA1, SR-B1, and other receptors in a similar way to apoA-I and for SAA-dependent reduction in the ability of HDL to promote cholesterol efflux and other steps of RCT [80,261,262].

In addition to apoA-I, displacement of other HDL constituents might be indirectly induced by SAA. SAA can displace important antioxidant enzymes PON1 and Lp-PLA2, diminishing antioxidative capacity and increasing the proatherogenic properties of HDL [267]. In contrast, a positive correlation between enhanced antioxidant activity of HDL and elevated SAA levels was observed *in vivo* and *in vitro* studies using recombinant human SAA and HDL samples from patients with inflammation, suggesting antioxidant properties of SAA [262,266]. Isolated or recombinant SAA was able to inhibit oxidative stress in human neutrophils and induce the anti-inflammatory cytokine IL-10 [268].

Besides the indirect effect of impairing antiatherogenic HDL function by structure remodelling, SAA can exert a direct activation of inflammation. Stimulation of VSMCs with human HDL-SAA significantly induced MCP-1 mRNA expression. SAA itself can induce a wide array of cytokines and chemokines *via* binding to cell surface receptors and activating associated (receptor-dependent) signalling pathways [269]. Several structurally diverse receptors, including formyl-peptide receptor-like 1 (FPR2), SR-B1, TLR2 and TLR4, CD36, the receptor for advanced glycation end product (RAGE), were determined [260,269]. SAA-mediated FPR2 activation resulted in production of MCP-1 in VSMCs, monocytes, and ECs [20,269]. Synthesis and secretion of MCP-1 following SAA binding were shown to be dependent also on TLR2 and TLR4 in VSMCs and mice in a study using small interfering RNA (siRNA) technology, specific receptor agonists and antagonists, and special mouse strains [269]. SAA binding to TLR2 led to increased expression of pro-inflammatory interleukins (IL-6, IL-1, IL-33) and anti-apoptotic genes *via* NF-κB activation [260]. In *in vitro* experiments with HDL from healthy human subjects supplemented with SAA in human aortic endothelial cells (HAECs), impaired vascular functions of HDL (characterised by NO and ROS production, VCAM-1 expression, and endothelial mononuclear cell adhesion) were estimated [261].

In contrast, *in vitro* studies revealed that the ability of SAA to stimulate IL-1β secretion in macrophages (via Nod-like-receptor protein 3 (NLRP3)) was abolished upon associating SAA to HDL [270]. Many effects attributed to SAA were reported to be lost upon binding of SAA to HDL. Therefore, some authors suggest that the aim of SAA association with HDL is to reduce pro-inflammatory activities of lipid-poor SAA in the circulation and to transport inactive SAA to the site of inflammation, where it can be released to fulfil its role in innate immunity during APR [262,266]. Future studies are needed to elucidate the role of SAA enriched HDL in chronic inflammation and to distinguish the effects of systemic liver-derived and locally produced SAA to evolve functioning strategies of therapy.

### 6.18. Ceruloplasmin

Ceruloplasmin is an acute phase protein, working as a plasma copper carrier with ferric oxidase activity. During APR, ceruloplasmin increases in HDL (but not in LDL or VLDL) of human patients [22]. The serum concentration of ceruloplasmin positively correlates with TAGs, age, and BMI, it is increased in subjects with MetS, T2DM, and CAD [271]. The association of ceruloplasmin with CVD is not fully understood. Ceruloplasmin inhibits the oxidation of lipids and prevents protein and DNA damage. An antioxidant effect of ceruloplasmin’s ferroxidase activity and inhibition of the Fenton reaction, which uses Fe^2+^ to generate ROS, was proposed. However, under oxidative stress, copper can be released from the molecule, promoting vasculopathic effects, working pro-oxidant as it increases lipid oxidation, in particular oxidation of LDL, apoptosis of ECs, and lowers bioavailability of NO [272].

### 6.19. Fibrinogen

Fibrinogen is a complex glycoprotein playing a central role in haemostasis, where soluble fibrinogen is polymerised into a non-soluble fibrin polymer to form a fibrin clot. The fibrin formation is tightly regulated *via* the fibrinolytic system, preventing aberrant clot formation. Besides that, fibrinogen binds and activates platelet and also binds to many other endogenous molecules, like von Willebrand factor, fibronectin, albumin, vascular endothelial growth factor, thrombospondin, fibulin, fibroblast growth factor-2 or IL-1 [273]. Whereas fibrinogen levels markedly increase after injury or infection during the inflammatory response, fibrinogen is considered a positive acute phase reactant. Since fibrinogen and some molecules created during its degradation or polymerisation participate in the regulation of inflammatory response, its role in inflammatory response has become the subject of great scientific interest. For example, fibrinopeptide B, generated during fibrinogen polymerisation, may act as a chemoattractant for leukocytes, and fibrinogen can activate immune cells or NF-κB pathway *via* CD11b/CD18 signalling. The effects on inflammation, platelet aggregation, plaque composition, or blood viscosity predispose fibrinogen to participate in atherosclerosis pathology [274]. Several studies suggest that increased fibrinogen levels correlate with the presence and severity of atherosclerosis [275,276,277,278] and a specific fibrinogen phenotype is associated with increased atherosclerosis [279]. Enrichment of synovial fluid with fibrinogen-derived citrullinated proteins [280] was observed in RA patients. A long-term cohort study identified fibrinogen as the independent marker of CVD, but the incidence of myocardial infarction and death, predicted by plasma levels of fibrinogen, is modified by its covariance with other inflammation-sensitive proteins [281].

### 6.20. Hp, Hb and Hx

Hp and Hx are liver-derived proteins with protective roles against harmful effects of excessive free Hb under haemolysis. Iron contained in Hb can participate in the Fenton reaction producing ROS that can damage proteins, lipids, and nucleic acids. The detoxication of Hb is mediated by binding of Hb to Hp and Hx, respectively, and clearance of complexes by anti-inflammatory routes mainly to the liver macrophages using specific receptors (CD163 for Hp, CD91 for Hx). The synthesis of Hp and Hx is induced by various cytokines involved in the inflammatory process [282]. Elevated Hb concentration in plasma of CHD patients was determined. It was demonstrated that the association of a complex of Hb.Hp.Hx with apoA-I containing HDL positively correlated with inflammatory properties of HDL in patients with CHD and in mouse models of hyperlipidaemia [283]. Hb bound in HDL can cause overconsumption of antioxidants [284] and oxidative modification of HDL-associated proteins, such as apoA-I, PON1, and Gpx-3 [285], impairing HDL functions. Results of experiments with Hp^−/−^ and Hx^−/−^ models of mice suggest Hp pro-inflammatory and Hx anti-inflammatory roles regarding HDL function [283]. In addition, a functional polymorphism at the Hp gene locus was shown to be associated with marked differences in HDL structure and function, such as the impaired ability of HDL to promote cholesterol efflux from macrophages, and elevated prevalence of coronary endothelial dysfunction (an early stage of atherosclerosis), especially among DM individuals. It was shown that increased HDL-bound Hb content, which can be 5x higher in DM individuals, resulted in impaired vascular protective effects of HDL due to the sequestering of NO and thus in decreased bioavailability of NO. Especially, DM patients with Hp2-2 genotype exhibit a significant increase in HDL-bound Hb and coronary endothelial dysfunction [286].

### 6.21. AAT

**AAT** is a positive acute phase reactant synthesized mainly in the liver, protectiing body tissues against proteases produced by immune cells during the immune response. It is a member of the serine proteinase inhibitor (serpin) family. According to recent studies, AAT could be characterised also as an anti-inflammatory and immuno-regulatory protein which interferes also with lipid metabolism through increased expression of LPL inhibitor, angiopoietin-like 4 [287]. HDL enrichment with AAT protected HDL particles from elastase-mediated loss of CEC and free-cholesterol esterifying functionality [288], and HDL-AAT complex reduced bronchial emphysema and inflammation more efficiently than HDL or AAT alone [289]. According to some studies, low serum levels of AAT and variations of *AAT* gene are associated with atherosclerosis progression [290,291].

### 6.22. α1-Acid Glycoprotein (AAG)

AAG, or orosomucoid is another positive acute phase reactant synthesised mainly in the liver. It works as a modulator of immune response and transporter protein, influences vascular permeability, and possesses some anti-inflammatory properties [292]. Association of elevated AAG with carotid plaques, the incidence of ischemic stroke [293], and in type 1 diabetic patients with endothelial dysfunction and subclinical atherosclerosis [294] were observed.

### 6.23. α2-Macroglobulin (α2M)

α2M is a proteinase inhibitor synthesised in the liver. By inactivation of proteases, it influences a variety of physiological functions (e.g., inhibition of fibrinolytic system by plasmin inhibition, inhibition of coagulation by thrombin inhibition) [295], suppresses the activity of pro-inflammatory cytokines, regulates the expression of a wide spectrum of genes involved in the proliferation of cells, oncogenesis, and atherosclerosis. The specific circulating molecular form is probably involved in myocardial infarction and cardiac hypertrophy pathogenesis [296]. Some studies suggest that high plasma levels of α2M are associated with increased atherosclerotic plaque vulnerability [297] and that CVD risk correlates with increased α2M content in HDL [298].

### 6.24. β2-Microglobulin (B2M)

B2M is a small protein that represents a structural component of the major histocompatibility complex (MHC) class I molecule, thus participating in the immune response. It is present in many body fluids and on almost every nucleated cell [299]. Informations about the association of B2M with inflammation and CVD risk are controversial. Although some studies show an increase in B2M levels during exacerbation of inflammatory diseases [300,301] suggesting its potential as a marker of inflammatory disease activity, others found no correlation with markers of systemic inflammation [302,303]. The association between higher B2M levels and cardiovascular outcome in patients with atherosclerosis [304], and with coronary and peripheral artery disease [305,306] were observed. On the contrary, the study in chronic haemodialysis patients found decreased mortality in patients with higher B2M levels and a negative correlation of B2M levels with HDLc [307].

### 6.25. Secretory Phospholipase A2 (sPLA2)

sPLA2 is an esterase associated with HDL catalysing the hydrolysis of glycerophospholipids to FFAs and lysophospholipids. Released lysophospholipids are precursors for the PAF and, besides that, during this process also the most important pro-inflammatory fatty acid, arachidonic acid, could be produced. The role of arachidonic acid in inflammation and its metabolism to pro-inflammatory mediators, eicosanoids, are well known [308]. However, as the group V sPLA2 isoform seems to exert anti-inflammatory potential in the study of Boilar et al., this suggests that the overall effect of sPLA2s on inflammation depends on specific sPLA2 isoform [309]. Increased sPLA2 expression in mice led to decreased HDL and apoA-I levels, changes in HDL content (depleted phospholipids and CEs, increased proteins and TAGs), and accelerated HDL catabolism [310,311], and in humans and animal models correlates with atherosclerosis development [312,313,314].

### 6.26. Complement Component 3 (C3)

C3 plays a crucial role in innate immunity as a mediator of complement activation. All three pathways of complement activation, classical, alternative, and lectin, converge in C3 activation and the C3 component subsequently interacts with other complement components and non-complement proteins, thereby rendering important immune function in defence against pathogens [315]. C3 could bind to HDL particles [97] and increased levels of C3 seems to be associated with atherosclerosis [316,317].

## 7. Conversion of Anti-Inflammatory HDL to piHDL and Its Relevance to Atherosclerosis

As mentioned before, in the basal state, HDL possesses many protective activities, like anti-inflammatory, antioxidative and anti-atherogenic. During chronic inflammation, markedly reduced biogenesis and increased catabolism of HDL lead to low HDLc in plasma. In addition, structural alterations, such as (i) replacing of typical HDL-associated proteins with protective activities (e.g., apoA-I, PON-1) by pro-inflammatory proteins (e.g., ceruloplasmin or SAA), (ii) post-translational modifications of proteins (e.g., oxidation, carbamylation, glycation), (iii) enrichment of TAGs and depletion of CEs in hydrophobic core and (iv) alterations in phospholipids (due to changes in CETP, LPL, LCAT, sPLA2, Lp-PLA2 activities) occur [12,267,318,319,320,321,322]. These changes significantly influence the stability, functionality, and metabolism of HDL. Virtually all beneficial HDL functions, such as RCT, anti-inflammatory activity, and the ability to prevent oxLDL formation, can be diminished in piHDL under chronic stress conditions. During acute inflammation, as piHDL is probably able to accelerate the immune response, the short-term HDL remodelling could be a physiological part of the innate immune system with essential role in host-response against pathogens [17]. However, in chronic inflammatory diseases, like RA or psoriasis, the long-term remodelling of HDL particles is associated with accelerated atherosclerosis [323]. piHDL isolated from SLE patients directly influenced the chemotaxis of monocytes and upregulated production of pro-inflammatory molecules TNFα and MCP-1 [324], suggesting a role of piHDL in plaque initiation. Studying the correlation between HDL proteome and atherosclerosis progression, Gordon et al. revealed a positive association between calcified plaque burden and apoA-IV and apoC-II and a negative association with PON1. On the other hand, non-calcified plaque burden correlated negatively with apoA-I, apoF and apoC-I and positively with SAA and complement C3 component [325]. Proteomic analysis comparing HDL of healthy subjects and HDL of patients with CAD (HDLcad) by using liquid chromatography-electrospray ionization/multistage mass spectrometry revealed reduced apoJ and increased apoC-III HDL content, accompanied by the endothelial proapoptotic process in patients with stable and acute CAD [75]. It was suggested that decreased apoJ and increased apoC-III on HDLcad were factors affecting the regulation of endothelial apoptosis. HDLcad did not activate endothelial anti-apoptotic pathways, but rather stimulated proapoptotic pathway *via* p38 mitogen-activated protein kinase (p38 MAPK) activation. However, the vascular impact of HDL was found to be very variable in CAD patients [253].

## 8. Alterations of HDL Proteome under Inflammatory Conditions

Numerous animal or human studies were performed to discover the whole spectrum of changes in HDL proteome during inflammation. The results are consistent for some proteins. Increased SAA, ceruloplasmin, fibrinogen, Hp, C3 complement component, and decreased apoA-I or PON1 are almost constantly seen across animal as well as human studies. Inconsistencies in the quantification of other proteins could be the result of specific proteome changes under different pathological conditions, e.g., acute/chronic inflammation, specific disease [321], specific setting of the animal model, or human study. The time factor could also play a role (e.g., in the study of Van Lenten et al., increased apoJ returns to basal state 3 days after inflammatory stimulus) [326]. The quantification of plasma activity of Lp-PLA2 in some studies and isolation of HDL and quantification of only HDL-associated activity in others complicates the interpretation of results [327,328,329,330,331,332]. Moreover, during pathological conditions, the proteomic alternations could vary between specific HDL subfractions (e.g., increase in apoA-II in HDL2 and decrease in HDL3 in CAD) [115].

Estimated changes in HDL composition during acute or chronic inflammation gained from proteomic analysis of isolated HDL in animal and human studies are summarised in Table 1. Enzymatic activities of HDL-related enzymes are summarised in Table 2.

There are several studies, focusing on HDL changes in acute inflammation. Vaisar et al. studied the remodelling of HDL proteome in healthy subjects after endotoxin administration at a concentration of 1–2 ng/kg. They observed only a tiny remodelling of HDL composition and increased SAA was the only significant change. Since some HDL components may change (increase or decrease) proportionally to the degree of inflammation, the lack of significant changes may be explained by only mild inflammation induced by endotoxin in studied subjects [338]. Moya et al. in a human experimental study using endotoxin at a higher concentration than Vaisar (3 ng/kg) observed decreased plasma LCAT activity [351]. Decreased LCAT and CETP activities were observed also in the plasma of patients with acute sepsis by Reisinger et al. [340]. Interesting results are gained from animal studies. Decreased LCAT activity associated with decreased cholesterol removal from cells was observed in HDL isolated from LPS stimulated Syrian hamster [350]. Vaisar et al. observed massive HDL remodelling in C57BL/6 mice induced by subcutaneous injection of silver nitrate. Proteomic analysis of HDL composition showed decreased apoA-I, apoA-II, apoC-I, apoC-III, PON1, Hb, PLTP, AAT, B2M and increased fibrinogen, Hp, Hx, SAA, LBP, apoA-IV, apoE and AAG content in HDL of inflamed mice [338]. Van Lenten et al. tested the effect of croton oil-induced inflammation on HDL composition and function in New Zealand rabbits. They observed lower apoA-I content, PON1 and Lp-PLA2 activities, and higher ceruloplasmin and SAA. The same changes were observed also in humans with acute inflammation response after cardiac surgery [22]. In another experiment on C57BL/6 mice with acute influenza infection, this team discovered HDL enrichment with apoJ, ceruloplasmin, and decreased HDL-associated PON1, and Lp-PLA2 activities [326].

RA is an example of a chronic inflammatory disease with the long-term remodelling of HDL proteome. Charles-Schoeman et al. observed a significant decrease in LCAT plasma activity and increase in HDL-associated SAA concentration in patients with RA compared to healthy controls [106]. Increased CETP and decreased PON1 activities were observed in HDL isolated from patients with RA by Kim et al. [341]. Besides the HDL, decreased PON1 activity along with decreased LCAT activity were observed also in the plasma of RA patients [352]. Additionally, increased plasma activity of sPLA_2_ was observed [379] and the increase in activity probably correlates with disease severity [380]. Conflicting results may be seen in studies for plasma Lp-PLA2 activity [331,332], Gpx-3 activity [359,360,361]. In comparison to previous studies, Watanabe et al. compared the protein composition of HDL in rheumatoid patients with anti-inflammatory vs. pro-inflammatory HDL phenotype. According to them, RA patients with piHDL have significantly increased apoJ, fibrinogen, AAT, C3 complement component, Hp, and SAA. A noticeable but insignificant trend was seen in decreased apoA-I, apoA-II, apoC-I, PON1, and increased Hx, AAG [9]. These modifications in RA patients could lead to HDL dysfunction, reduced CEC, and thus enhance atherosclerosis development [381].

SLE is a chronic inflammatory diagnosis with autoimmune aetiology. Similarly to RA, dysregulation of lipid metabolism and remodelling of HDL proteome occurs in SLE. Parra et al. observed a significant change of 17 HDL-associated proteins in SLE patients compared to healthy control. Among other things, they found reduced apoA-I, AAG, α2M and transferrin, and enhanced apoM, apoC-I, apoC-II, apoC-III, and apoA-II in HDL of SLE patients [333]. Besides that, other studies found also reduced apoA-I and increased apoE content in HDL [339] and in serum of SLE patients, decreased PON1 [342], Gpx-3 activities [362,363,364] and increased Lp-PLA2 [328], sPLA_2_ [379] and CETP activities [342]. Like in RA patients, HDL in SLE has highly altered functionality and loses its function to prevent LDL oxidation. This may predispose the patients to atherosclerosis and explain increased CVD risk in these patients [323].

Psoriasis, another chronic inflammatory autoimmune disease, is also typical for increased risk of cardiovascular complications [382]. Similarly, as in previous diseases, this risk could be accelerated by pro-inflammatory HDL remodelling. In psoriatic patients, Holzer et al. observed significantly reduced apoA-I and apoM levels whereas apoA-II, SAA, transferrin, complement C3, AAT, AAG, Hb, and its scavenger protein Hp were enriched in psoriatic HDL. They also found increased Lp-PLA_2_ activity in HDL. Moreover, quantification of some proteins like fibrinogen, Hx, or apoA-IV has a non-significant but distinct increasing trend and apoC-I decreasing trend in psoriatic patients [330]. Other studies observed enhanced HDL-associated Lp-PLA_2_ activity, decreased PON1 activity in HDL [329], but also in plasma [353], and decreased serum Gpx-3 [365], LCAT [353,354], and CETP activity [343] in psoriatic patients compared to healthy subjects.

Chronic PD, manifesting as a long-term inflammation of gums, may also lead to increased CVD risk [383,384]. Dysregulation of lipid profile due to prolonged inflammation is a frequent characteristic seen in these patients [385,386]. Patients with PD exhibited an altered plasma lipoprotein profile (such as decreased apoA-I), correlating with inflammation-related proteins (IL-21, IL-17F, IL-7, IL-1RA, IL-2). Proteomic profiling of HDL isolated from PD patients showed significantly reduced levels of apoJ, PLTP, C3, and PON3, and increased apoA-II and apoC-III in HDL in comparison to healthy subjects. Non-significant trends were observed in increased apoA-I and decreased LCAT or AAT [5]. Other studies found reduced serum PON1 activity [374] and inconsistent results for Gpx3 activity [366,367,368].

Chronic inflammation plays a crucial role also in the pathogenesis of MetS and DM. HDL enriched with apoC-II, apoC-III, fibrinogen, Hx, SAA, and transferrin and depleted in apoJ, apoC-I, PON-1, C3, apoA-IV, PLTP, AAG, B2M, apoE and apoM was found in diabetic patients in a study by Cardner et al. However, according to this study, the role of specific HDL-proteomic alternations in coronary atherosclerosis development is not clear, mainly when considering the different remodelling pattern observed in CAD patients, although they have some similarities like HDL deprivation of apoA-IV and enrichment with pulmonary surfactant protein B and SAA [321]. Informations about plasma enzymatic activities in T2DM patients are sometimes controversial. In studies, we may found increased [355] but also decreased plasma LCAT activity in diabetic patients [356]. Similar controversies are found also in studies of Gpx3 activity [369,370]. According to some studies, CETP activity seems to be unchanged in T2DM [355,387,388], others suggest slightly decreased activity [344,345]. Decreased PON1 activity was observed in HDL [375], but also in plasma [376,377] of T2M patients. A decrease in CEC was reported in several studies with DM and MetS [389,390]. In patients with MetS, reduction in SR-B1 and ABCG1-dependent efflux correlated with the increase in individual criteria of MetS [391]. Decreased serum CEC in T2DM was found to be associated with endothelial dysfunction [392].

Compared to healthy subjects, patients with CKD are more prone to develop serious CVD events. This phenomenon is probably the result of complex alternations in body functions. There is a strong evidence that lipid metabolism is heavily dysregulated in CKD patients and that HDL undergoes modifications that transform it into a pro-inflammatory phenotype in CKD patients [393]. Using mass spectroscopy, Shao et al. studied the HDL composition of patients in the final stage of CKD progress, end-stage renal disease (ESRD). They observed that the HDL of these patients was significantly enriched with SAA1, apoC-II, apoC-III, apoA-IV, apoE, AAT, B2M, AAG and depleted of apoA-I, apoA-II, apoM, C3, PON1 and LCAT. A non-significant increasing trend was observed for Hp [334]. These results are relatively consistent with the findings of Holzer et al. [335]. However, compared to a study by Mangé et al., in which authors applied isobaric tags for relative and absolute quantitation (iTRAQ) labelling and nanoflow liquid chromatography-mass spectrometry analysis, some inconsistencies and differences could be found, such as increased apoA-II, apoC-I, LCAT, apoM, PON1 and decreased fibrinogen, apoJ, transferrin, Hp, Hx, α2M in CKD [336]. The variation in obtained results from different studies may be due to different comorbidities in CKD patients like DM [394]. A complex review of HDL proteomic studies is summarised elsewhere [25]. Besides that, in plasma of patients with CKD, increased CETP [346], Lp-PLA_2_ [349], decreased LCAT [357], Gpx-3 [371,372] and PON1 [346,378] activities were observed.

In the plasma of early NAFLD patients, lower apoA-I, lower preβ1-HDL, increased CETP activity, and no differences in LCAT and PLTP were determined [347,348]. Interestingly, decreased ABCA1-mediated CEC of NAFLD patient’s plasma in cyclic adenosine monophosphate (cAMP)-treated J774 cells (murine macrophages), but not ABCG1-dependent CEC in THP-1 cells, was found to be an independently associated factor with subclinical atherosclerosis (presence of carotid atherosclerotic plaque) [348]. Kasumov discovered that in NAFLD patients, HDL was enriched with complement C3 and ceruloplasmin, and depleted of apoA-II and PON1 [337]. Other studies in NAFLD patients found increased plasma LCAT [358], Gpx3 [373], and decreased PON1 activity [347]

To summarise, patients with chronic inflammatory diseases have an increased risk of cardiovascular comorbidities and atherosclerosis development. HDL from patients with atherogenesis-associated diseases share some common properties with HDL from patients with chronic inflammatory conditions. Understanding these changes and their role in the pathogenesis of atherosclerosis could be a key factor for future treatment.

## 9. Post-Translational Changes of HDL Proteins

Besides altered protein enrichment in piHDL in patients with chronic inflammatory diseases, several post-translational changes, such as oxidation, carbamylation, and glycation of HDL proteins and lipids, impairing atheroprotective properties of HDL, were reported [267,319,320,395].

### 9.1. Oxidation

Oxidation is one of the most important pathological modifications of proteins occurring in an inflammatory environment. ROS, produced *in vivo* in numerous types of cells, including immune cells or ECs, participate in redox signalling under physiological state, regulating cell proliferation and growth. However, under inflammatory conditions, the imbalance between the generation of ROS and antioxidant systems occurs, leading to oxidative stress. ROS can directly oxidise and damage proteins, lipids, and DNA and increase the production of pro-inflammatory mediators *via* NF-κB, activator protein 1 (AP-1), and other ROS-regulated transcription factors [396]. Numerous studies reported an increase in biomarkers of oxidative protein, lipid, and nucleic acid damage in patients with T2DM, RA, or PD indicating the oxidative damage of lipoproteins [5,319,320]. Components of piHDL are markedly oxidised, altering their structure and functions. Oxidised HDL (oxHDL) is found in atherosclerotic plaques [397]. Potential candidates for the oxidation of HDL proteins linked to the progression of atherosclerosis are MPO [397,398] and superoxide anion generated during NOS uncoupling [399].

MPO is a heme enzyme, a component of the innate immune response, secreted by activated neutrophils and macrophages in the circulation and the vascular wall. MPO plasma levels correlate with the risk of atherosclerosis [400]. Levels of MPO can be increased in plasma, but it can also be accumulated in the subintimal space of the artery wall leading to increased levels of oxLDL [69]. Besides binding to apoB in LDL, MPO associates with HDL in atherosclerotic lesions [23]. MPO can utilize NO, nitrite (NO_2_^−^), hypochlorous acid (HOCl), and other co-substrates with hydrogen peroxide (H_2_O_2_) to cause ROS promoting protein oxidation (halogenation, nitration, oxidative cross-linking), and lipid-peroxidation [401,402]. Mass spectrometry (MS) revealed elevated levels of 2 characteristic products of MPO, 3-chlorotyrosine, and 3-nitrotyrosine, in HDL from patients with CAD and CVD. MPO-dependent methionine oxidation and chlorination of a single tyrosine residue led to the loss of activity of apoA-I. As a result, numerous functions of HDL facilitated by apoA-I are affected, such as the ability of HDL to interact with ABCA1 to drive the cholesterol efflux from lipid-laden macrophages, to interact with PON1, or to activate LCAT to enable the maturation of HDL [115,285]. MPO-catalysed HDL oxidation results in loss of anti-apoptotic and anti-inflammatory functions of HDL dependent on SR-B1 binding; and activation of pro-inflammatory signalling cascades in ECs (NF-κB activation, VCAM-1 upregulation) [403]. PON1 as the third partner in a ternary complex of MPO with apoA-I in HDL has reduced activity due to oxidative modification of tyrosine 71 [69]. A decreased antioxidative PON1 activity and increased MPO activity led to significantly increased levels of oxidised lipids in HDL. Histidine and lysine residues of apoA-I, apoA-II, apoA-IV and apoE participate in cross-linking by oxidised phospholipids, as it was assessed *in vitro* and *in vivo* (in isolated human and murine HDL, or in plasma and aorta in mice, respectively), attenuating cholesterol efflux [398]. Selective MPO inhibitors or polymorphisms of the MPO gene are leading to decreased MPO activity, and reduced the risk or progression of atherosclerosis in apoE^−/−^ mice and humans, respectively [69].

### 9.2. Carbamylation

Several atherogenic effects, such as cholesterol accumulation, macrophage foam-cell formation, pro-inflammatory signalling, and vascular smooth muscle proliferation, are caused by carbamylated lipoproteins [267,404]. In human atherosclerotic lesions, HDL was shown to be a major target for carbamylation. Carbamylation of HDL proteins results in dysfunctional HDL particles. In human atherosclerotic lesions, amino acids of HDL proteins interact with isocyanate, a product of thiocyanate (major physiological substrate of MPO) oxidation by MPO [267]. MPO/HDL interaction increases upon oxidation of HDL but in the presence of thiocyanate MPO-induced protein carbamylation is favoured over chlorination (3-chlorotyrosine). Even a minimal carbamylation of apoA-I-HDL affects the interaction of HDL with SR-B1, destabilising the HDL/SR-B1 mediated balance between cholesterol-uptake versus efflux in macrophages, inducing cholesterol accumulation. The marked enrichment in the carbamyllysine content of apoA-I was detected in human atherosclerotic plaques. Since apoA-I is a major LCAT activator in HDL, apoA-I modification impairs LCAT functions in cholesterol esterification, maturation of HDL, and RCT. LCAT and PON1 were reported to be deactivated by carbamylation in a dose-dependent manner, and the antioxidative function of HDL was compromised, resulting in the reduced ability of HDL to inhibit radical-induced LDL oxidation *in vitro* [23]. The changed affinity of HDL to SR-B1 receptors may be the reason for the attenuated anti-apoptotic activity of HDL, as seen in induced endothelial cell death [23,405].

### 9.3. Glycation

Increased glycemia is accompanied by the formation of advanced glycation end products (AGEs) as a result of the non-enzymatic glycation of proteins or lipids in plasma (after exposure to sugars). The AGEs formation is stimulated also by oxidative stress. Glycation modifies protein function, disrupts its conformation, interferes with receptor binding, and thus may accelerate many pathological processes [406]. In diabetic patients, hyperglycaemia induces glycation of many HDL components, like apoA-I and transferrin, and deamidation of ceruloplasmin [407]. Glycated apoA-I has a shorter half-life [408], impaired ability to promote cholesterol efflux and stabilise ABCA1, and reduced cardioprotective and antiatherogenic properties [409]. Conformational change in glycated apoA-I significantly affects the ability of apoA-I to bind lipids and activate LCAT, or CETP and PON1 [267]. Glycation of PON-1 decreased its enzymatic activity and attenuated its protective activity against monocyte adhesion to aortic epithelial cells *in vitro* [410]. Glycated apoA-IV lost its ability to reverse the LPS-induced changes in cholesterol efflux-associated gene expression [122]. In HUVEC cells, AGEs also inhibit the expression of SR-B1, the receptor responsible for cholesterol uptake [411]. It was clinically observed that glycation of apoA-I together with decreased PON activity correlated with the incidence and severity of CAD in diabetic patients [26]. Glycated HDL-impaired cholesterol efflux *in vitro* was shown to be reverted by metformin, a widely used anti-diabetic agent [412]. The AGEs formation could be seen not only in DM but also in other diseases. Highly glycated and probably dysfunctional HDL particles were observed in RA [319,341]. According to a study in SLE patients, these patients have a higher formation of AGEs compared to control, and AGEs accumulation might contribute to atherosclerosis development [413]. Increased AGEs formation was further seen in CKD [395,414] or NAFLD [415]. Taken together, the formation of AGEs could impair HDL’s anti-atherogenic [416], antioxidant, and anti-inflammatory [417] properties and is associated with the formation of atherosclerotic lesions and cardiovascular complications [267].

## 10. Currently Available HDL-Targeted Therapies

In general, lifestyle modification including diet, exercise, weight loss, and smoking cessation, is recommended for increasing HDL levels and improving HDL function in patients with increased CVD risk. In patients with chronic low-grade inflammation, effective disease control may attenuate CVD risk. As atherosclerosis is an inflammatory process, an important target of therapy in patients is the decrease in the level of systemic inflammation [418]. Long-term intensive medical therapy combined with lifestyle modification lowered not only weight, fasting glucose, and inflammatory markers (CRP, TNFα) in T2DM patients, but also improved the anti-oxidative function of HDL [419]. Effective periodontal treatment reduced markers of systemic inflammation, increased HDLc, improved endothelial function, and favourably influenced the status of PD’s frequent comorbidities, T2DM, RA, or NAFLD/NASH [7]. Studies endorsed the increase in all lipoproteins including HDL by disease-modifying antirheumatic drugs (DMARDs) or biological therapy in RA, correlating with the decline of CRP levels. The increase in lipoproteins reflects the normalisation of the lipid levels, which are not increasing CVD risk, but only improving the „lipid paradox “seen in RA [8,106]. The treatment of RA patients with methotrexate exhibited a significant decrease in CVD-associated mortality in comparison with other DMARDs [420]. In patients with autoimmune rheumatoid diseases RA and SLE, anti-inflammatory biologic drugs, such as anti-TNF, anti-IL6, or anti-IL1β agents, can decelerate the atherogenic process [76]. The decrease in inflammation (estimated by the CRP level) reached by therapies of RA with abatacept or adalimumab improvement in inflammation and disease activity, correlated with improvement of antioxidant HDL functions and alterations of HDL proteome (LBP, SAA-I, Hp, PON1 activity) [421]. However, although patients in remission have better parameters than those with active disease, their level of inflammation and ratio of apoB/apoA-I lipoproteins might be still higher than those of healthy subjects, representing increased CVD risk in comparison with the general population. The primary anti-inflammatory therapy is therefore often accompanied by additional lipid-modifying agents [422]. Recently, there have been only a limited number of drugs in clinical practice specifically affecting HDL particles, among them fibrates, niacin, or statins. In addition, the results of clinical trials of currently approved agents are sometimes controversial, thus their efficacy in improving CVD risk is questionable and the therapy is often accompanied by unwanted side effects. Taken together, this highlights the necessity for finding new, more effective drugs for cardiovascular application.

### 10.1. Drugs in Clinical Practice

**Fibrates** are well-known and widely used agents in clinical practice, which work mainly as synthetic ligands for PPARs (PPARα, PPARγ, PPARδ), transcription factors regulating lipid metabolism and inflammation. After activation by fibrates, PPARs bind to PPAR response elements in promoter regions of target genes and modulate their expression. Fibrates increase lipolysis *via* activation of lipoprotein lipase, suppression of apoC-III synthesis, and upregulation of expression of genes involved in the β-oxidation pathway. They also induce hepatic fatty acid uptake through increased expression of fatty acid transporter protein (FATP) and acyl-CoA synthetase [423] or increase gene and protein expression of apoA-I and apoA-II, thus enhancing HDL formation. Moreover, through induction of LXRα, ABCA1, ABCG1, and SR-B1 expression, they accelerate RCT. Activation of PPARα also reduces the number of pro-atherogenic small dense LDL particles [424]. Besides direct effects on lipid metabolism, fibrate’s anti-atherogenic activity could be explained also by the inhibition of inflammatory conditions in artery walls. For example, they suppress monocyte adhesion on artery walls probably through inhibition of the pro-inflammatory transcription factor NK-ĸB, thereby inhibiting VCAM-1 expression [425]. Reduction in inflammatory markers and thrombin production in patients treated with fenofibrate demonstrates fibrates’ anti-inflammatory, but also anti-thrombotic activity. The incidence of serious CVD events, such as stroke or CHD, was not influenced by fibrate treatment according to some studies, however, a positive effect on non-fatal coronary events was observed [426,427]. Another study with bezafibrate shows an effective reduction in fatal or nonfatal myocardial infarction or sudden death compared to placebo mainly in a subgroup of patients with high baseline TAG levels. [428]. Fenofibrate addition to statin therapy in patients with MetS shows promising effectiveness in the reduction in CVD events [429]. Gemfibrozil treatment was associated with reduced CVD events in patients with low HDL levels [430]. In another study, fibrates prolonged survival in diabetic patients with dyslipidaemia [431]. A positive effect of fenofibrate on dysregulated lipid profile in RA patients was observed [432], however large randomized controlled studies are needed to confirm these results. Promising outcomes of the studies are confirmed by meta-analyses, which show fibrates’ effectivity, mainly in patients with specific metabolic conditions like metabolic syndrome, high TAG levels, or atherogenic dyslipidaemia profile [433,434,435,436]. Rhabdomyolysis and hepatic diseases, the most serious side effects of fibrates, are the main concern for caution when prescribing these drugs by physicians [437]. Pemafibrate, a potent and selective PPARα modulator, was shown to perform cardiovascular beneficial functions of other known fibrates, but without off-target effects on liver and kidney health. Based on the clinical data of better risk-benefit balance and no drug–drug interactions of pemafibrate with statins, it may find application for patients with a variety of metabolic diseases, including hyperlipidaemia, CHD, T2DM, NAFLD/NASH, CKD [424,438,439,440]. In 2017 pemafibrate gained its first global approval in Japan for the treatment of hyperlipidaemia (including familial hyperlipidaemia) [441]. On the other side, a large European clinical trial PROMINENT, evaluating pemafibrate’s efficacy in reducing cardiovascular outcomes in diabetic patients was prematurely discontinued due to the improbability of achieving its primary endpoint [442].

**Nicotinic acid**, also known as **niacin**, is one of the group B vitamins. Supplementation with niacin increases HDLc *via* enhanced hepatic apoA-I production and inhibition of SR-B1-independent HDL-holoparticle uptake and catabolism in the liver [443,444]. Besides the significant increase in HDL and decrease in TAG levels that were clinically observed in patients, in many trials was the niacin therapy unsuccessful in decreasing CVD events [445,446]. This was confirmed also by the COCHRANE meta-analysis observing that niacin did not reduce the number of deaths, heart attacks, or strokes. Moreover, a high rate (18%) of discontinuation of the treatment due to the side effects (e.g., skin flushing, headache, itching…) was observed [447]. On the other side, a meta-analysis from Hongwei et al. reported reduced carotid IMT in patients treated with the combination of statins and niacin versus statins alone or in combination with ezetimibe [448].

The efficacy of **statins** on CVD risk is mainly based on their ability to decrease plasmatic levels of LDL. This effect is mediated by competitive inhibition of 3-hydroxy-3-methylglutaryl coenzyme A (HMG-CoA) reductase, a key enzyme in cholesterol biosynthesis. As a consequence of low hepatic cholesterol levels, increased expression of LDL receptors on the surface of hepatocytes leads to enhanced clearance of LDL from circulation. Moreover, decreased secretion of VLDL due to inhibition of apoB100 synthesis was reported. Although their effect on HDL cholesterol levels is not as strong as for LDL, it could be beneficial [449,450,451]. The increase in HDLc may be mediated by the activation of PPARα and an increase in hepatic ApoA-I mRNA expression or through CETP inhibition leading to reduced exchange of CEs from HDL to VLDL [452,453]. Besides metabolic activity, statins also possess a wide spectrum of pleiotropic effects. Among other mechanisms, for statins’ anti-inflammatory, antioxidant, anti-atherogenic and immune-modulatory activity are responsible: the down-regulation of pro-inflammatory transcriptional factors as NF-ĸB or AP-1, and decreased production of inflammatory mediators, such as IL-1β, TNFα, IL-6, IL-8, MCP-1 and CRP, inhibition of T-lymphocytes activation, modulation of NO cascade, lowering (nicotinamide adenine dinucleotide phosphate) NADP oxidase, superoxide formation and increased scavenging of free oxygen radicals [454,455]. Notably, statins differ markedly in their potency to influence these parameters [456]. The positive effect of statins on CVD risk was observed also in clinical trials in patients with specific inflammatory conditions, like MetS [457], DM [458], SLE [459] or RA [460,461] and statins’ beneficial effect on CVD was confirmed by meta-analyses [462,463]. Therapeutic usage of statins is complicated by their adverse events, e.g., raised incidence of DM and cataracts, and frequent muscular side effects [464].

The strategies for the treatment of dyslipidaemia could be based also on **antisense oligonucleotide** technology. By hybridization with specific mRNA molecules, antisense oligonucleotides may affect the expression of specific genes or inactivate miRNAs involved in HDL structure, metabolism, and function. For example, in 2019, the Europe medicine agency (EMA) approved **volanesorsen**, an **antisense oligonucleotide against ApoC-III mRNA**, which prevents the synthesis of apoC-III protein as a treatment for familial chylomicronaemia syndrome [465]. Since some studies suggest that apoC-III-bound lipoproteins have altered metabolism and apoC-III is involved in many pro-atherogenic processes, inhibition of the synthesis could have protective activity [466]. The authorization was preceded by clinical trials APPROACH and COMPASS, where among others, an increase in plasmatic HDL and a decrease in TAGs and non-HDL levels were observed [467,468,469]. Positive effects on HDL levels, accompanied by improved insulin sensitivity and decreased TAGs, were confirmed also in the Phase II study in patients with DM [470].

Mechanisms of action of these agents are schematically depicted in Figure 1 and the characteristics of the drugs are summarised in Table 3.

### 10.2. HDL-Affecting Drugs in Clinical Trials for Cardiovascular Application

Intense scientific research focusing on HDL brought many promising therapeutic agents in past few years. A more precise understanding of HDL role and function allows the development of molecules improving HDL functionality by targeting specific HDL components. Many of these agents confirmed their efficacy in clinical trials, being strong candidates for future approval. However, in some cases, the conclusions of pre-clinical experiments did not correspond with clinical results. HDL-affecting molecules, achieving the level of clinical trials, are summarised in the following subchapter.

The development of **CETP inhibitors** was driven by the observation of the lipid profile of patients with genetic CETP deficiency, which typically have elevated HDL levels whereas their LDL levels are reduced. Based on the hypothesis, the pharmacological inhibition of CETP was supposed to reduce atherosclerosis *via* increased HDL and decreased LDL levels [471]. Unfortunately, despite the clinical confirmation of CETP inhibitors’ effect on HDL and LDL levels in patients, dalcetrapib [472] and evacetrapib [473] failed to decrease CVD risk in clinical trials. Furthermore, torcetrapib increased the CVD event and risk of mortality and morbidity in patients. The increase may be explained by the off-target effects of this agent [474]. A recent study, evaluating the data from anacetrapib and torcetrapib clinical trials, brought another explanation for the suboptimal clinical efficacy of these agents. Even though in these trials increased apoA-I content of HDL particles was observed, mainly the specific HDL subspecies associated with increased coronary risk (for example apoC-III containing HDL) were enriched with apoA-I and preferentially the plasmatic concentrations of such HDL subspecies were upregulated. Thus, the lipoprotein profile was shifted towards the type associated with increased CVD risk [475]. Considering that, the development of these agents for cardiovascular application was interrupted. Another CETP inhibitor, anacetrapib, improved RCT, plasma levels and, anti-oxidative capacity of HDL in transgenic mice, but an unwanted acceleration of fatty liver and insulin resistance were observed [476]. Although the 4-year long clinical trial of anacetrapib in combination with statins showed a lower incidence of major coronary events [477], this effect was not sufficient for approval and anacetrapib development was abandoned by the manufacturer due to an inappropriate clinical profile in 2017 [478]. However, recent observations from the longer follow-up of the study participants showed an increased therapeutic effect of anacetrapib treatment after 6 years period with favourable safety profile. These results indicate that the development of this drug might have been abandoned prematurely and for proper evaluation of the safety and efficacy of lipid-modifying drugs, longer treatment and follow-up periods in clinical trials are needed [479]. Besides synthetic molecules, the activity of CETP may be inhibited also by monoclonal antibodies [480] and the special vaccination approach, eliciting anti-CETP antibodies in order to reduce its activity, was also tested in clinical trials [481,482,483]. Clinical inefficiency of many aforementioned agents which increase HDL levels led to a modification of the general therapeutic strategy. From the efforts of increasing HDL quantity, the researchers focused their attention on the improvement of its qualitative characteristics. As a part of this strategy, **small peptides mimicking apoA-I**, which promote the formation of pre-β HDLs, increase cholesterol efflux, suppress inflammation, and prevent lipoprotein oxidation, were developed [484,485,486]. The most studied apo-AI mimetic peptide is 4F. It is produced in versions with all D-amino acids (D-4F) and all L-amino acids (L-4F). The ability of D-4F to reduce the pro-inflammatory properties of HDL was clinically confirmed in high CVD risk patients [487], whereas L-4F seemed to be clinically ineffective [488]. Moreover, next apoA-I mimetics with better characteristics were developed, like more ABCA1-specific and less cytotoxic 5A or LCAT-specifically activating ETC-642 [489]. Promising pre-clinical results, a good safety profile, but insufficient clinical efficacy for apoA-I mimetics were summarised in a meta-analysis by Abudukeremu et al. [490]. The **reconstituted HDL particles** are the complex of recombinant apoA-I with phospholipids which mimic pre-βHDL particles. Their atheroprotective effect may be due to the promotion of cholesterol efflux from lipid-enriched macrophages [491]. Despite a significant reduction in atherosclerotic plaques by reconstituted HDL complex ETC-216 in patients with acute coronary syndrome [492], its development was interrupted due to its serious adverse immunostimulation. As a consequence, modified MDCO-216 lacking the immune-stimulating effect was introduced [493], however, this molecule did not prove its efficacy [494]. CER-001 reported improved CEC along with beneficial effects on inflammatory activity and plaque burden in the aorta and carotid arteries of patients with dyslipidaemia in small human studies. Nevertheless, in the larger randomized clinical trial, significant plaque regression was not observed [495]. Another reconstituted HDL CSL-111 significantly improved the plaque characterization index and coronary score in patients. However, it did not affect atheroma volume and also caused an increase in transaminases [496]. Thus, its modified second-generation version, CSL-112, with a better safety profile is under investigation [497].

Other clinically tested proteins which mimic physiological functions of some HDL components are **apoE mimetic peptides** [137]. Studies suggest that apoE plays a role in the physiological repair mechanism of HDL particles disabled by the excessive amounts of cholesterol. The enrichment of the particles with apoE may improve their CEC and RCT. The great advantage of allowing apoE to clear excess cholesterol more efficiently than other apolipoproteins is based on the fact that it could clear plasmatic cholesterol through SR-B1 but also LDLR. This hypothesis is supported by the results of a cohort study, where the higher ratio of apoE in HDL decreased the CVD risk in patients with cholesterol-overloaded HDL particles [498]. ApoE mimetics shows also anti-inflammatory, anti-atherogenic properties, and the ability to reduce plasmatic cholesterol levels in animal models [499,500,501,502]. ApoE mimetic peptide AEM-28 was promisingly tested in Phase I/II clinical trial as a treatment for refractory hypercholesterolemia, where a reduction in the total amount of VLDL and TAGs after AEM-28 treatment was observed [503,504,505]. A peptide encoding the putative LDLR binding sequence of apoE linked to 18A synthetic peptide similar to apoA-I, designated as Ac-hE18A-NH_2_, was able to mediate cholesterol efflux, reduced plasma cholesterol and increased PON1 activity, and reduced SAA in animal models. Ac-hE18A-NH_2_ was found to be more effective in anti-inflammatory and anti-atherogenic action than 4F [499].

Another potential molecule for improving HDL quality is **monoclonal antibody** MEDI5884. The antibody is targeted against **endothelial lipase** (EL), the phospholipase responsible for HDL metabolism. EL promotes lipid depletion, destabilization, and renal clearance of HDL particles [506]. *In vivo* experiments in mice revealed that antibody-mediated inhibition of EL led to increased HDLc and the size of HDL particles [507]. A positive effect of MEDI5884 on HDL was confirmed in non-human primates, but also in a human Phase I study on healthy volunteers, where MEDI5884 was able to increase HDLc, average HDL size, improved cholesterol efflux, and anti-inflammatory properties of HDL [506]. Similar encouraging results were seen also in Phase IIa study on patients with stable coronary artery disease [508].

Epigenetics, one of the fastest developing fields of interest in biology, provides another treatment option for improving cardiovascular health. RVX-208 or apabetalone belongs to so-called **bromodomain and extra-terminal domain (BET) inhibitors**. BET proteins are epigenetic readers which influence gene transcription by recognising and binding to acetylated lysine in histones. BET inhibitors prevent this regulation by disrupting of connection between BET proteins and lysine [509]. Apabetalone selectively inhibits BET protein 4 (BRD4), which increases gene and protein apoA-I expression and CEC [510], and reduces vascular inflammation by suppressing inflammatory and adhesion molecule gene expression [511]. The positive effects of apabetalone on cardiovascular systems, like a significant increase in HDL, apoA-I mRNA and protein, large HDL particles and average HDL particle size [510], modulation of plaque vulnerability [512], and decreased probability of heart failure in patients with T2DM with acute coronary syndrome [513] were confirmed in Phase II and III clinical trials. Further clinical evaluation of apabetalone’s efficacy is still ongoing.

LCAT, an enzyme activated by apoA-I, catalyses the esterification of cholesterol in HDL. Since the migration of nascent CEs to the interior of HDL subsequently promotes cholesterol efflux and drives HDL maturation from discoidal pre-β HDL to spherical α-HDL, it is considered a critical step of RCT [514]. Given that, administration of recombinant LCAT may lead to increased cholesterol esterification and enhanced RCT. Although some animal studies suggest that the **overexpression of LCAT** does not result in enhanced RCT [239,515], in clinical trials in patients with CHD, administration of the **recombinant LCAT** (ACP-501, another name MEDI6012) increased HDLc, favourably altered HDL metabolism, increased the apoA-I levels, CEC and raised the number of large HDL particles whilst small and intermediate-size HDL particles along with proatherogenic LDL particles decreased [516,517]. Many **small-molecule LCAT activators**, such as DS-8190a, have been successfully tested for the reduction in atherosclerotic lesions in animal models [518]. DS-8190a was proved to be able to activate not only wild-type but also some naturally occurring human LCAT mutants *in vitro* [519]. Compound A ((3-(5-(ethylthio)-1,3,4-thiadiazol-2-ylthio) pyrazine-2-carbonitrile) increased HDLc, apoA-I, HDL particle size and decreased VLDL and TAGs in mice and hamsters [520] (still in preclinical phase).

LXRs are transcription factors playing important role in lipid metabolism and inflammation. As sensors for oxidized sterols, LXRs maintain cholesterol homeostasis by directly and indirectly regulating genes involved in cholesterol efflux, transport, intestinal absorption, and excretion, influencing multiple steps of RCT [521,522]. **LXR agonists** are a novel treatment option. Molecular mechanisms include LXR-mediated increase in ABCA1, ABCG1 in intestinal enterocytes, hepatocytes and macrophages, and ApoA-I mRNA expression in the intestine, leading to an increase in HDL levels and improved RCT [521,523]. In studies, these effects were confirmed in healthy individuals. On the other hand, LXR agonists stimulate hepatic fatty acid synthesis *via* enhanced expression of sterol regulatory element-binding protein-1c (SREBP-1c). In animal models and humans, CNS-associated side effects and risk of fatty liver and steatosis were observed [524,525].

A cannabinoid receptor 1(CB1) influences appetitive behaviour and energy metabolism, and therefore it was proposed to be a target for anti-obesity treatment. An **inverse agonist of CB1**, Rimonabant, decreased the size and severity of atherosclerotic lesions in mice. A positive anti-atherogenic effect was observed despite rimonabant attenuating RCT in this animal model. The effect may be explained by modulating lipid profile *via* decreased VLDL-TAG production and accelerated VLDL-TAG turnover, decreased LDL levels, and increased HDL levels [526]. The study in abdominally obese patients with atherogenic dyslipidaemia showed significant changes in HDL particle sizes, HDL2 and HDL3 amount, increased apoA-I, and decreased apoB and CRP levels [527]. However, the treatment was associated with serious psychiatric side effects [528] and further use of this drug in clinical practice in Europe was prohibited soon after marketing authorisation by EMA [529]. The United States Food and Drug Administration (FDA) never approved this drug for use in The United States of America [530].

Even though Lp-PLA2 was for a long time considered an enzyme with anti-atherogenic potential, recent evidence suggesting its association with increased risk for atherosclerosis [208] led to the development of a new class of anti-atherogenic therapy, **Lp-PLA2 inhibitors**. Although oral Lp-PLA2 inhibitor darapladib showed promising anti-inflammatory and plaque-reducing properties in a pre-clinical swine model of atherosclerosis [531] and subsequent clinical trial, it prevented the expansion of necrotic core that is the determinant for plaque vulnerability, in large clinical trials STABILITY and SOLID TIMI 52, darapladib failed to improve the incidence of CVD events in patients [532,533]. Insufficient penetration of darapladib into plaques could be a possible explanation for its clinical failure. Since the majority of serum Lp-PLA2 is bound to LDL particles, the higher affinity of darapladib to HDL may provide an additional explanation for its failure. In case the hypothesis of different activity of HDL or LDL-bound Lp-PLA2 is a true, a more detailed understanding of this phenomenon could also help to clarify darapladib inefficiency [534].

Recent studies suggest that also many naturally occurring compounds may be promising therapeutic options for the improvement of CVD risk. Antioxidant and anti-inflammatory effects of **flavonoid and non-flavonoid polyphenols** are well-documented [535]. Mainly in patients with chronic inflammatory conditions, polyphenols could improve HDL function, enhance RCT or protect HDL particles against dysfunction. The precise mechanism for the protective activity varies depending on the specific chemical structure of concrete polyphenol. In general, the mechanisms may include: (i) activation of transcriptional factors, such as LXR or PPAR, (ii) regulation of ABCA1, ABCG1, SR-B1 expression, (iii) increasing plasmatic levels of apoA-I or HDL, (iiii) increasing activity of anti-oxidant/anti-atherogenic enzymes (PON1, LCAT, …) [536]. For instance, citrus bioflavonoid **hesperidin**, which naturally occurs in citrus fruit, exerts anti-atherogenic potential probably through the improvement of RCT by up-regulation of ABCA1, ABCG1, or increased levels of antioxidant enzymes, which was observed in LDLR-deficient (LDLR^−/−^) mice [537] A plant flavonol **quercetin** reduced atherosclerotic lesions and altered ABCA1, LXR-α, and PCSK9 expression in apoE-deficient (apoE^−/−^) mice [538] or increased PON1 activity in Wistar rats [539]. Flavanone **naringenin** improved atherosclerosis, increased HDL and Gpx-3 levels, and promoted cell autophagy in high-fat-diet (HFD)-fed apoE^−/−^ mice. Anti-atherogenic effect of green tea **catechins** belonging to the flavanols subgroup could be mediated by upregulation of ABCA1, ABCG1, and SR-B1 expression [540] or increased PON1 activity, as observed in rat models [541]. **Anthocyanins**, natural pigments belonging to flavonoids, increased PON1 activity and enhanced cholesterol efflux in apoE^−/−^ mice and could be another potential anti-atherogenic plant derivate [542]. **Resveratrol** inhibited LDL and HDL oxidation, enhanced apoA-I mediated cholesterol efflux by up-regulating ABCA1 and reduced cholesterol influx in macrophages [543], ameliorated atherosclerosis induced by high-fat diet (HFD), or LPS in apoE^−/−^ mice [544]; and prevented the development of atherosclerotic lesion, lowered the Lp-PLA2 levels in HFD-fed rabbits [545]. **Paeonol**, a phenolic compound found in *Paeonia suffruticosa*, retarded atherosclerosis progression in apoE^−/−^ mice by upregulation of ABCA1-mediated enhancement of cholesterol efflux and downregulation of CD36 [546]. Polyphenol **curcumin** reduced atherosclerotic lesions *via* its anti-inflammatory effect in LDLR^−/−^ mice, increased liver ApoA-I mRNA, PPARα and LXRα expression, plasma HDLc and lowered CETP activity [547]. Effects of curcumin supplementation from pre-clinical models were tested in clinical studies in diabetic patients [548,549], NAFLD patients [550], or healthy volunteers [551] in which some positive influence on CVD risk factors was observed. Many of the aforementioned positive effects of polyphenols on HDL particles were confirmed in clinical settings, like increased HDL and antioxidant functions in subjects with T2DM or improved cholesterol efflux, inhibited CETP, and increased HDL levels in dyslipidemic subjects by anthocyanins supplementation [552,553,554], increased PON1 activity by catechins-supplementation in haemodialysis patients [555], increased HDL cholesterol by flavanones in orange juice [556] and others [536]. A positive effect of greater polyphenol intake on CVD risk was confirmed also in the PREDIMED study [557].

From **other**
**plant metabolites and components**, potentially effective could be pseudoalkaloid **leonurine,** which upregulated PPARγ, LXRα, and ABCA1/G1 expressions, improved plasma lipid profile, and decreased the size of atherosclerotic lesions in apoE^−/−^ mice [558]. The major active principle of garlic, **allicin**, is an organosulphur compound with many biological effects. At first, it was considered an antibacterial agent, and later studies revealed also its possible hypoglycaemic, hypolipidemic [559], antioxidant [560], antihypertensive, cardioprotectant [561], or antifungal [562] activities. The positive effect on cardiovascular health could be associated also with PPARγ/LXRα-mediated upregulation of ABCA1 expression thus improving cholesterol efflux, observed *in vitro* [563] but also in an animal model [564]. Other *in vivo* studies observed an increase in HDL-associated antioxidant Gpx-3 [565], reduction in atherosclerotic plaques, and positive lipoprotein modifications [566]. A positive effect of allicin supplementation was observed also in humans [567]. Supplementation of a carotenoid obtained in tomatoes, **lycopene,** in overweight middle-aged individuals led to decreased SAA, increased PON1 and LCAT activity in specific HDL subfractions and serum, whilst decreasing serum CETP activity [568]. The meta-analysis of available trials revealed its positive effects on blood lipids, blood pressure and endothelial function [569]. ***Gymnema Sylvestre* extract**, whose active compounds are so-called gymnemic acids, is a part of Indian traditional medicine studied mainly due to its anti-diabetic properties. The animal experiments show also an effective increase in serum apoA-I, HDL levels along with an increase in liver levels of some HDL-associated antioxidant enzymes, such as Gpx-3 [570] or an increase in blood LCAT activity in HFD-fed rats [571]. Increased intake of non-digestible plant material in the **dietary fibre** could also have a beneficial effect on atherosclerosis development and cardiovascular health [572]. Among others, the observed increase in HDL cholesterol and HDL to TC levels ratio by fibre from animal [573] and human [574] studies could contribute to this protective activity. The effect of more nutraceuticals on atherosclerosis in numerous cohort studies, clinical trials, and epidemiological studies was in more detail summarised elsewhere [575].

In general, no benefit of antioxidant **vitamins C** or **E** supplementation on CVD risk was observed in most clinical studies, but these vitamins can be helpful under special circumstances [575]. Increased association of Hb with HDL, observed in DM individuals with Hp2-2 genotype, is proposed to result in the oxidation of HDL and the diminishing of antioxidants, such as vitamin E [284,286]. Administration of antioxidant **vitamin E** to diabetic individuals with Hp2-2, but not with Hp1-1, reversed HDL dysfunction (CEC) and HDL-bound peroxidation [576].

The beneficial effect of omega-3 and omega-6 polyunsaturated fatty acids (**PUFAs**), ligands for PPARs, on major CVD-related events (0–19% in clinical trials) is linked mostly to their TAG-, VLDL- and oxLDL-lowering effect [575]. Interestingly, a small pilot study with obese volunteers revealed that high content of dietary unsaturated fat triggered acceleration of RCT steps specifically on apoE-containing HDL: apoE-HDL synthesis, size expansion, cholesterol uptake by the liver, and holoparticle clearance [577], which might improve cholesterol balance in the arterial wall. In another study, a 12-week program of therapeutic lifestyle change, comprising a Mediterranean diet (rich in PUFAs) and exercise in patients with MetS, improved CEC along with the decrease in MPO-oxidation products: 3-chlorotyrosine and 3-nitrotyrosine [578].

Since the dysbiosis of gut microbiota or chronic bacterial infection of the oral mucosa are contributing inflammatory factors for the development of atherosclerosis, an increased interest has been attracted to the effect of **probiotics.** The intestinal microbiota regulates various functions of the host by maintaining the integrity of the intestinal barrier, stimulating the growth or activity of beneficial bacteria, and eliminating pathogens and production of trimethylamine N-oxide, providing a series of metabolites, such as short chain fatty acids; affecting numerous signalling pathways including those involved in lipid metabolism, atherosclerosis, and inflammation [579]. Microbial metabolites, MAMPs, and PAMPs, and the corresponding pathways could play a role in the inhibition of inflammation, influencing potentially HDL’s nature and the process of atherosclerosis [580]. Probiotics were shown to be able to inhibit the production of pro-inflammatory cytokines (TNFα, IL-6, IL-8) by the attenuation of TLR pro-inflammatory cascades in numerous studies [51,581]. *Lactobacillus (L.) casei* supplementation improved inflammatory status (decreased CRP, IL-6, TNFα, and increased IL-10) and alleviated disease activity in RA patients [582,583]. The normalisation of gut microbiota by probiotic or prebiotic supplementation might be helpful in the treatment of NAFLD, IBD, hyperlipidaemia, obesity, and T2DM [51,580,584,585]. Probiotic mixture VSL#3 [586] and *L. rhamnosus* GG [587] reduced lesion development in HFD-fed apoE^−/−^ mice by attenuation of vascular inflammation and by altering the colonic gut microbiota, respectively. The meta-analysis of randomised clinical trials revealed a significant increase in HDLc, and a significant decrease in CRP in T2DM patients following supplementation with probiotics [588]. Usually, a multispecies probiotic or symbiotic supplementation was shown to be more potent than single-species supplementation [588,589,590]. A consumption of symbiotic food, consisting of *L. sporogenes* and inulin, resulted in a significant reduction in TAGs, VLDL, and in a significant increase in HDLc in patients with T2DM in a small, randomized double-blinded controlled clinical trial [591]. Additionally, in *L. acidophilus*-treated apoE^−/−^ mice under HFD, the development of atherosclerotic lesions was prevented by the inhibition of lipid accumulation [592]. Reduction in TC, potentially acting to prevent and ameliorate obesity and hyperlipidaemia [585], is explained by the inhibiting effect of probiotics and synbiotics on cholesterol absorption in the intestine and inhibition of HMG-CoA reductase, but mechanisms of increasing HDLc and conferring the anti-atherogenic effect of probiotics remain to be revealed [588]. In *in vitro* experiments, *L. acidophilus* K301 upregulated mRNA and protein expression of LXR-dependent genes, ABCA1, and ABCG1, in phorbol 12-myristate 13-acetate (PMA)-differentiated THP-1 cells, resulting in increased apoA-I- dependent cholesterol efflux [592]. Supplementation of *L. acidophilus* CHO-220 with prebiotic inulin in hypercholesterolemic individuals for 12 weeks significantly increased the amount of CEs in HDL indicating potentiated LCAT activity [593]. The impact of probiotics on HDL’s quality, where a lower incidence of unwanted side effects is to be expected, represents an interesting and unexplored field of study.

Mechanisms of action of clinically tested agents are schematically depicted in Figure 2 and characteristics of the drugs are summarised in Table 4.

### 10.3. Experimental Therapies

Besides many molecules whose development reached the level of large clinical trials, there are many other compounds tested as HDL-improving therapies in experimental animal models and/or in small pilot studies. The oral administration of **apoJ mimetic peptide** D-[113–122]apoJ: (i) increased cholesterol efflux from macrophages *via* the ABCA1 pathway in apoE^−/−^ mice, (ii) reduced lipoprotein lipid hydroperoxides in monkeys, (iii) increased PON1 activity and rendered anti-inflammatory properties of HDL in both, mice and monkey models [594]. Moreover, D-[113–122] apoJ reduced atherosclerosis in apoE^−/−^ mice and LDLR^−/−^ mice, improved the antioxidant function of HDL, and its function to promote RCT in LDLR^−/−^ mice [594,595].

The development of **apoC-II mimetics** was driven by the observation that administration of apoA-I mimetics could cause transient hypertriglyceridemia, probably *via* decreasing the activity of LPL. The main advantage of apoC mimetics, which are able to promote cholesterol efflux by ABCA1 similarly to apoA-I, is that they also promote the LPL-mediated lipolysis and therefore lack this unwanted hypertriglyceridemic effect of apoA-I mimetics. The ABCA1-mediated effect on cholesterol was confirmed *in vitro* [596] and the TAG-lowering effect also in animal models [597].

The previously mentioned agent, Volanesorsen, may not be in the future the only antisense oligonucleotide used for the treatment of atherosclerosis. Many oligonucleotides, targeted against other RNAs associated with HDL function, were tested in experimental models. For example, **antisense oligonucleotides against CETP mRNA** increased HDLc and improved RCT in hyperlipidemic mice model [598]. In addition, **antisense oligonucleotide-mediated inhibition of miR-33a/b** shows promising results in pre-clinical settings. The physiologic functions of miR-33a/b include among others the suppression of ABCA1 expression, the key regulator of HDL biogenesis, or affecting the expression of genes associated with fatty acid metabolism. Thus, inhibiting these miRNAs by antisense oligonucleotides, may increase hepatic ABCA1 expression and HDL production and lower TAG levels. This effect was observed in primate [599] and mice [600] models.

Targeting SR-B1 was considered one of the potential strategies for improving cardiovascular health. It was assumed that inhibition of SR-B1-mediated uptake of HDL to the liver would have an anti-atherogenic effect by increasing plasmatic HDLc and decreasing its clearance. Although an expected increase in HDL and apoA-I was observed clinically in hypertriglyceridemic subjects treated with **SR-B1 inhibitor** ITX5061 [601], further study of its atheroprotective effect did not proceed. Recently, the drug development of ITX5061 transferred from cardiovascular application to antiviral application as a treatment against hepatitis C [602].

It was discovered that increased intake of **hydrogen (H_2_)** through inhalation or water enriched with H_2_ has potent antioxidative activity. Such an effect could be potentially used for improving many medical conditions as well as for improving lipid profile and function. The positive effect of H_2_-enriched water on lipoproteins, such as increased protective effect against LDL oxidation, increased cholesterol efflux, inhibition of monocyte adhesion to ECs, and susceptibility of ECs to apoptosis, was confirmed not only in an animal model but also in a small patient study [603]. Additionally, H_2_ treatment decreased Lp-PLA2 and increased PON1 and LCAT activity in HFD-fed rats [604].

*In vivo* studies reveal that the purified **sardine proteins** improved RCT through increased activity of LCAT and reduced HDL oxidation by increasing the activity of PON1 in rats fed a cholesterol-rich diet [605,606].

A potential treatment strategy for atherosclerosis is also **photodynamic therapy** with photosensitisers. Photosensitisers selectively accumulate within atherosclerotic plaques and after activation by light, they trigger different death pathways, resulting in reduction in plaque burden [607]. Besides the direct effect of photodynamic therapy on proliferating cells, *in vitro*, it was observed that silica upconversion nanoparticles encapsulating photosensitiser chlorin e6 (Ce6) in combination with a 980 nm laser also increased ABCA1-mediated cholesterol efflux in THP-1-derived foam cells. This effect was probably mediated by the induction of autophagy through the ROS/PI3K/Akt/mTOR signalling pathway [608,609]. A positive effect of different formulas of Ce6 photosensitiser on atheroma plaques, which may be partially explained by increased cholesterol efflux, was observed also *in vivo* in mice [610] and rabbits [611].

Mechanisms of action of experimental drugs are schematically depicted in Figure 3 and characteristics of the agents are summarised in Table 5.

## 11. Conclusions

Patients with chronic inflammatory diseases, like RA, SLE, psoriasis, T2DM, CKD, NAFLD, or PD, are at higher risk of developing cardiovascular complications.

The mechanism by which inflammation increases CVD risk is likely multifactorial but HDL cholesterol may play a significant role. During chronic inflammation, changes in composition and thus conversion of anti-inflammatory HDL particles into pro-inflammatory occur. This process is characterised by changes in protein components (decreased apoA-I, PON1, increased SAA, etc.), and posttranslational modifications (oxidation, carbamylation, or glycosylation) of HDL particles that attenuate their anti-atherogenic, antioxidant, or anti-inflammatory properties. Although the precise mechanisms and physiological functions of HDL remodelling are currently not well understood, it is assumed that such modifications may play a role in the innate immune system. However, a prolonged persistence of converted HDL particles in an organism could lead to many unwanted side effects and promote atherosclerosis.

This might also explain why, despite the higher HDL levels are in general considered anti-atherogenic, in patients with chronic inflammation, therapeutic approaches focused on raising HDL cholesterol levels failed to decrease CVD risk. Recently, it became evident that instead of simply increasing the HDLc, improving HDL composition and functionality could be more meaningful targets for therapy. As a consequence, many therapies trying to improve HDL functionality are under development. Precise studying of changes in HDL structure and function during inflammatory conditions, together with a better understanding of the mechanism of HDL remodelling, may provide valuable knowledge for the improvement of cardiovascular health in patients with chronic inflammatory diseases.

## Figures and Tables

**Figure 1 pharmaceuticals-15-01278-f001:**
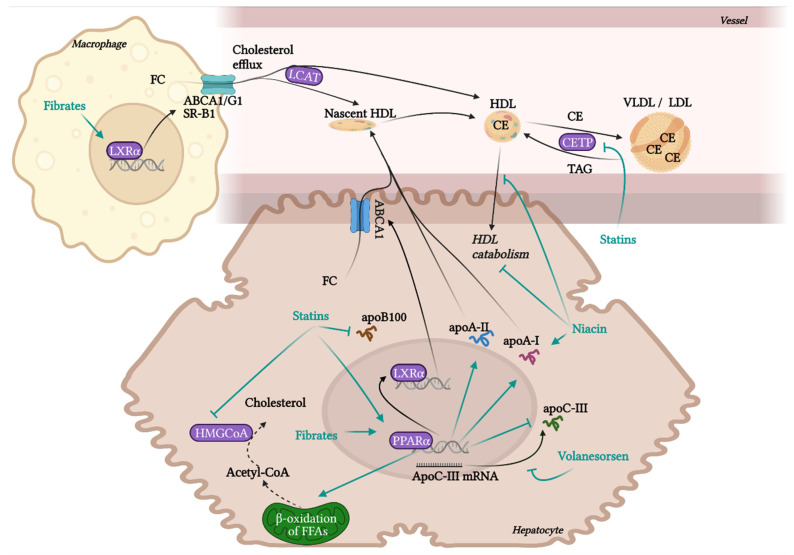
Simplified schematic representation of HDL-targeted therapies: Drugs in clinical practice. Apolipoprotein A-I (apoA-I); apolipoprotein A-II (apoA-II); Apolipoprotein B100 (apoB100); Apolipoprotein C-III (apoC-III); ATP-binding Cassette Receptor A1 (ABCA1); ATP-binding cassette receptor A1/G1 (ABCA1/G1); Cholesterol ester (CE); Cholesteryl ester transfer protein (CETP); Free Fatty Acids (FFAs); Free Cholesterol (FC); Hepatic scavenger receptor B1 (SR-B1); Lecithin-Cholesterol Acyltransferase (LCAT); Liver X Receptor α (LXRα); Peroxisome Proliferator-Activated Receptor α (PPARα); Triacylglycerols (TAG). Created with BioRender.com (accessed on 15 September 2022).

**Figure 2 pharmaceuticals-15-01278-f002:**
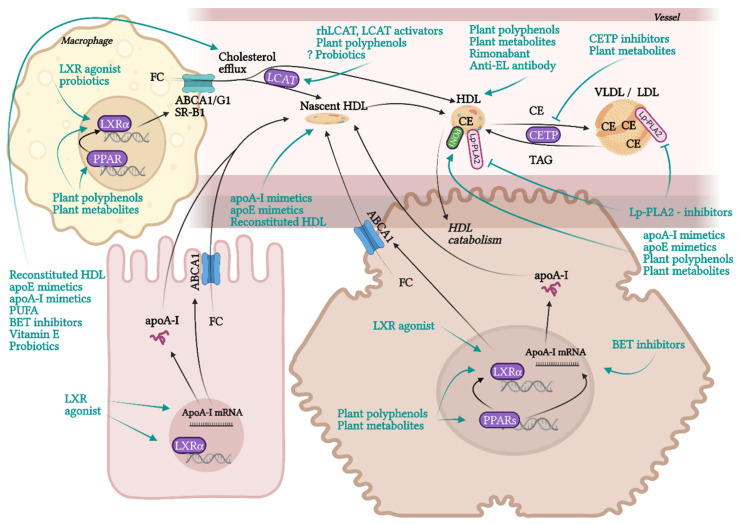
Simplified schematic representation of HDL-targeted therapies: Drugs in clinical trials. Apolipoprotein A-I (apoA-I); ATP-binding Cassette Receptor A1 (ABCA1); ATP-binding cassette receptor A1/G1 (ABCA1/G1); Cholesterol ester (CE); Cholesteryl ester transfer protein (CETP); Free Cholesterol (FC); Hepatic scavenger receptor B1 (SR-B1); Lecithin-Cholesterol Acyltransferase (LCAT); Lipoprotein-associated phospholipase A2 (Lp-PLA2); Liver X Receptor α (LXRα); Paraoxonase 1 (PON1); Peroxisome Proliferator-Activated Receptor (PPAR); Triacylglycerols (TAG). Created with BioRender.com (accessed on 15 September 2022).

**Figure 3 pharmaceuticals-15-01278-f003:**
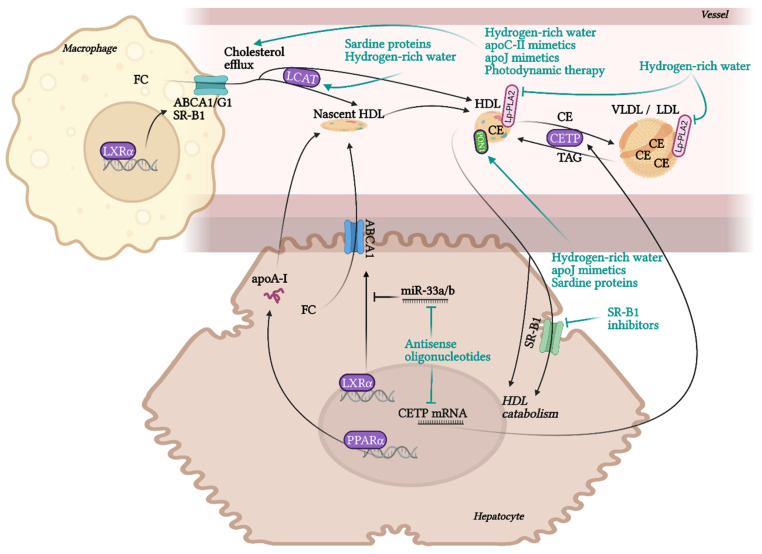
Simplified schematic representation of HDL-targeted therapies: Experimental drugs. Apolipoprotein A-I (apoA-I); ATP-binding Cassette Receptor A1 (ABCA1); ATP-binding cassette receptor A1/G1 (ABCA1/G1); Cholesterol ester (CE); Cholesteryl ester transfer protein (CETP); Free Cholesterol (FC); Free Fatty Acids (FFAs); Hepatic scavenger receptor B1 (SR-B1); Lecithin-Cholesterol Acyltransferase (LCAT); Lipoprotein-associated phospholipase A2 (Lp-PLA2); Liver X Receptor α (LXRα); Paraoxonase 1 (PON1); Perixosome Proliferator-Activated Receptor α (PPARα); Triacylglycerols (TAG). Created with BioRender.com (accessed on 15 September 2022).

**Table 1 pharmaceuticals-15-01278-t001:** Studies of HDL proteome in acute and chronic inflammatory conditions.

	Acute Inflammation	Chronic Inflammation									
		RA [9]	SLE [333]	Psoriasis [330]	PD [5]	T2DM [321]	CKD_1_ [334]	CKD_2_ [335]	CKD_3_ [336]	NAFLD [337]	Other
**apoA-I**	Mice ^1^: ↓ [338] Rab ^2^: ↓ [22] hSurg ^3^: ↓ [22]	↓ ^ns^	↓	↓	↑ ^ns^		↓	↓			SLE: ↓ [339]
**apoA-II**	Mice ^1^: ↓ [338]	↓ ^ns^	↑	↑	↑		↓	↓	↑	↓	
**apoA-IV**	Mice ^1^: ↑ [338]			↑ ^ns^		↓	↑	↑	↑		
**apoE**	Mice ^1^: ↑ [338]					↓	↑		↑		SLE: ↑ [339]
**apoC-I**	Mice ^1^: ↓ [338]	↓ ^ns^	↑	↓ ^ns^		↓		↓	↑		
**apoC-II**			↑			↑	↑	↑	↑		
**apoC-III**	Mice ^1^: ↓ [338]		↑		↑	↑	↑	↑	↑		
**apoJ**	Mice ^4^: ↑ [326]	↑			↓	↓			↓		
**apoM**			↑	↓		↓	↓	↓	↑		
**Lp-PLA2**								↑			
**PLTP**	Mice ^1^: ↓ [338]				↓	↓					
**LBP**	Mice ^1^: ↑ [338]										
**LCAT**					↓ ^ns^		↓		↑		
**PON1**	Mice ^1^: ↓ [338]	↓ ^ns^				↓	↓		↑	↓	
**Transferrin**			↓	↑		↑			↓		
**SAA**	Mice ^1^: ↑ [338] Rab ^2^: ↑ [22] hSurg ^3^: ↑ [22] hEtox ^5^: ↑ [338]	↑		↑		↑	↑	↑	↑		RA: ↑ [106]
**Ceruloplasmin**	Rab ^2^: ↑ [22] hSurg ^3^: ↑ [22] Mice ^4^: ↑ [326]									↑	
**Fibrinogen**	Mice ^1^: ↑ [338]	↑		↑ ^ns^		↑			↓		
**Hp**	Mice ^1^: ↑ [338]	↑		↑			↑ ^ns^		↓		
**Hb**	Mice ^1^: ↓ [338]			↑							
**Hx**	Mice ^1^: ↑ [338]	↑ ^ns^		↑ ^ns^		↑			↓		
**AAT**	Mice ^1^: ↓ [338]	↑		↑	↓ ^ns^		↑	↑			
**AAG**	Mice ^1^: ↑ [338]	↑ ^ns^	↓	↑		↓	↑				
**α2M**			↓						↓		
**B2M**	Mice ^1^: ↓ [338]					↓	↑		↑		
**C3 complement**		↑		↑	↓	↓	↓		↓	↑	

^ns^ non-significant trend (*p* < 0.15). ^1^ silver nitrate induced inflammation in C57BL/6 mice. ^2^ croton oil-induced inflammation in New Zealand rabbit model. ^3^ acute inflammation after cardiac surgery in humans. ^4^ influenza induced inflammation in C57BL/6 mice. ^5^ endotoxaemia in humans.

**Table 2 pharmaceuticals-15-01278-t002:** Enzymatic activities of HDL-related enzymes measured in plasma or HDL.

Enzyme	Acute Inflammation	Chronic Inflammation
**CETP**	hSeps ^1^: ↓ ^pl^ [340]	RA: ↑ ^HDL^ [341] SLE: ↑ ^pl^ [342] Psoriasis: ↓ ^pl^ [343] T2DM: ↓ ^pl^ [344,345] CKD: ↑ ^pl^ [346] NAFLD: ↑ ^pl^ [347,348]
**Lp-PLA_2_**	Rab ^2^: ↓ ^HDL^ [22] hSurg ^3^: ↓ ^HDL^ [22] Mice ^4^: ↓ ^HDL^ [326]	RA: ↓ ^pl^ [331] ↑ ^pl^ [332] SLE ^7^: ↑ ^pl^ [328] Psoriasis: ↑ ^HDL^ [329,330] CKD: ↑ ^pl^ [349]
**LCAT**	hSeps ^1^: ↓ ^pl^ [340] Ham ^5^: ↓ ^HDL^ [350] hEtox ^6^: ↓ ^pl^ [351]	RA: ↓ ^pl^ [106,352] Psoriasis: ↓ ^pl^ [353,354] T2DM: ↑ ^pl^ [355] ↓ ^pl^ [356] CKD: ↓ ^pl^ [357] NAFLD: ↑ ^pl^ [358]
**Gpx3**		RA: ↑ ^pl^ [359] ↓ ^pl^ [360,361] SLE: ↓ ^pl^ [362,363,364] Psoriasis: ↓ ^pl^ [365] PD: ↑ ^pl^ [366] ↓ ^pl^ [367,368] T2DM: ↑ ^pl^ [369] ↓ ^pl^ [370] CKD: ↓ ^pl^ [371,372] NAFLD: ↑ ^pl^ [373]
**PON1**	Rab ^2^: ↓ ^HDL^ [22] hSurg ^3^: ↓ ^HDL^ [22] Mice ^4^: ↓ ^HDL^ [326]	RA: ↓ ^HDL^ [341] ↓ ^pl^ [352] SLE: ↓ ^pl^ [342] Psoriasis: ↓ ^HDL^ [329] ↓ ^pl^ [353] PD: ↓ ^pl^ [374] T2DM: ↓ ^HDL^ [375] ↓ ^pl^ [376,377] CKD: ↓ ^pl^ [346,378] NAFLD: ↓ ^pl^ [347]
**sPLA_2_**		RA, SLE: ↑ ^pl^ [379]

^1^ sepsis in humans. ^2^ croton oil-induced enflammation in New Zealand rabbit model. ^3^ acute inflammation after cardiac surgery in humans. ^4^ influenza induced inflammation in C57BL/6 mice. ^5^ LPS induced inflammation in Syrian hamster. ^6^ endotoxaemia in humans. ^7^ only in SLE women with CVD versus SLE controls without CVD. ^pl^—activity measured in plasma. ^HDL^—activity measured in HDL.

**Table 3 pharmaceuticals-15-01278-t003:** HDL-affecting drugs used for cardiovascular application in clinical practice.

Drug	Effect	Additional Information	Reference
**Fibrates**	PPAR ligands, ↑apoA-I, ↑apoA-II → ↑HDL biogenesis, ↓apoC-III, ↑ABCA1/G1 and SR-B1 expression → ↑RCT	some trials showed lack of efficacy, better effect was observed in the combination with statins, recent meta-analyses showed promising reduction in CVD risk in population with specific metabolic conditions	[423]
**Niacin**	liver: ↑apoA-I, ↓HDL uptake, ↓HDL catabolism → ↑HDLc	trials show lack of benefit in patients with CVD concomitantly treated with statins	[443]
**Statins**	↑PPARα → ↑ApoA-I mRNA, ↓CETP	beneficial effect on CVD risk was confirmed in meta-analyses, effect on HDL is not the primary mechanism of statins’ therapy, but it could contribute to its efficacy, observed raised incidence of DM, muscular side effects	[452,463]
**Volanesorsen**	antisense oligonucleotid against ApoC-III mRNA → ↓apoC-III synthesis	increase in plasmatic HDL and decrease in TAGs and non-HDL levels was observed in clinical trials	[468]

**Table 4 pharmaceuticals-15-01278-t004:** HDL-affecting drugs for cardiovascular applications tested in clinical trials.

Drug	Effect	Additional Information	
**CETP inhibitors**	↓CETP → ↑HDLc	in most clinical trials, CETP-inhibitors failed to decrease CVD risk, trial with anacetrapib showed efficacy in reduction in coronary events but with side effects—liver steatosis, IR	[471]
**ApoA-I mimetics**	small peptides mimicking apoA-I → ↑pre-β HDLs, ↑CEC, anti-inflammatory properties of HDL	oral apoA-I mimetic D-4F was well tolerated and effectively reduced the inflammatory index of HDL particles in patients	[487]
**Reconstituted HDL**	recombinant apoA-I with phospholipids → ↑CEC	some molecules were clinically ineffective, some showed high incidence of adverse effects. Second-generation molecules with good safety profile are under investigation	[491,493,494,495]
**ApoE mimetics**	↑clearance of cholesterol by SR-B1 and LDLR → ↑CEC and RCT, ↑PON1, ↓SAA	AEM-28 was promisingly tested in Phase I/II clinical trial as treatment for refractory hypercholesterolemia	[503,504]
**Anti-EL monoclonal antibody**	↓EL → ↑HDLc, ↑HDL particle size, ↑cholesterol efflux, improved anti-inflammatory HDL function	MEDI5884 was succefully tested in Phase IIa study on patients with stable coronary artery disease	[508]
**Epigenetic therapy** (bromodomain and extraterminal domain (BET) inhibitor apabetalon)	↑ApoA-I mRNA/apoA-I expression, ↑CEC	clinical trials show effective modulation of lipid profile, improvement of vascular inflammation and reduction in CVD events in T2DM patients	[510,511]
**rhLCAT**	recombinant human LCAT → ↑formation of HDL and RCT	in patients with CHD, rhLCAT increased HDLc, favourably altered HDL metabolism and trial showed also good tolerability and safety of treatment	[516]
**LXR agonist**	↑ABCA1/G1, ↑ApoA-I mRNA, ↑RCT	the effect on ABCA1/ABCG1 expression was confirmed in a clinical trial, however, CNS-associated side effect and risk for steatosis was observed in animal models or humans	[523,524,525]
**Rimonabant**	inverse agonist of CB1 → ↑apoA-I, ↑HDL particles size, ↑HDLc	effect on HDL was confirmed in clinical practice, serious psychiatric side effects led to discontinuation of clinical usage	[527,528]
**Lp-PLA2-inhibitors**	↓Lp-PLA2	darapladib failed to reduce CVD events in patients	[532,533]
**Plant polyphenols**	LXR or PPAR activation, ↑apoA-I, ↑HDLc, ↑ABCA1/G1, SR-B1 → ↑RCT, ↑PON1, ↑LCAT and others…	positive effect on HDLc and properties was confirmed in many clinical trials	[536,546,547]
**Other plant metabolites and compounds**	LXR or PPARγ activation, ↑HDLc, ↓SAA, ↑ABCA1/G1, → ↑RCT ↓CETP, ↑Gpx-3, ↑PON1 and others…	positive effects on cardiovascular system were observed in many animal and human studies	[558,567]
**Vitamin E**	↑CEC, ↓HDL dysfunction and peroxidation	effect observed only in Hp 2-2 but not Hp1-1 T2DM individuals	[576]
**PUFAs**	↑RCT in apoE-HDL, ↓MPO oxidation products	Human	[577,578]
**Probiotics and synbiotics**	↑HDLc, ↑ABCA1/G1, ↑RCT, ↑CEs in HDL = ↑LCAT?	mice, human	[588,592,593]

**Table 5 pharmaceuticals-15-01278-t005:** Characteristics of experimental drugs affecting HDL.

Drug	Effect	Tested Subjects	
**ApoJ mimetic peptides**	↑RCT, ↓lipoprotein lipid peroxides, ↑PON1, improved anti-inflammatory HDL function	Mice, monkeys	[594,595]
**ApoC-II mimetic peptides**	↑ABCA1-mediated cholesterol efflux	mice	[596,597]
**Antisense oligonucleotides**	altering expression of specific genes (CETP)	mice	[598]
	inhibiting miRNAs involved in HDL metabolism and function (miR-33 a/b)	primate, mice	[599,600]
**SR-B1 inhibitors**	↓SR-B1-mediated uptake of HDL to liver → ↑HDLc	Humans, mice	[443,601]
**Hydrogen-rich water**	↓Lp-PLA2, ↑LCAT, antioxidant activity (↑PON1) → improved HDL function, ↑CEC	humans, rats, hamsters	[603,604]
**Purified sardine proteins**	↑LCAT → ↑RCT, ↑PON1 → ↓HDL oxidation	rats	[605,606]
**Photodynamic therapy with photosensitisers**	↑ABCA1-mediated cholesterol eflux	rabbits, mice	[610,611]

## Data Availability

Data sharing not applicable.

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
