# Peer review of "Alterations of HDL’s to piHDL’s Proteome in Patients with Chronic Inflammatory Diseases, and HDL-Targeted Therapies"

_pharmaceuticals, 2022, doi:10.3390/ph15101278_

Round 1
Reviewer 1 Report
Dear Authors, please describe one paragraph with the use of photodynamic therapy with photosynthetizers and/or use of antibody in described research.
Please use prisma analysis.
Sincerely
Author Response
Dear reviewer,
We would like to thank you for taking the time and effort necessary to review the manuscript. We appreciate your comments and suggestions, which have been helpful in improving the manuscript.
We revised our manuscript. Revisions are marked using “Track Changes” to easily view all changes.
Point 1: Dear Authors, please describe one paragraph with the use of photodynamic therapy with photosynthetizers and/or use of antibody in described research.
Answer 1: We added a paragraph about photosensitizers (pages 39-40, rows 1745-1755), and changed table 5, figure 3, and graphical abstract.
“A potential treatment strategy for atherosclerosis is also photodynamic therapy with photosensitisers. Photosensitisers selectively accumulate within atherosclerotic plaques and after activation by light, they trigger different death pathways, resulting in reduction of plaque burden [607]. Besides the direct effect of photodynamic therapy on proliferating cells, in vitro, it was observed that silica upconversion nanoparticles encapsulating photosensitiser chlorin e6 (Ce6) in combination with a 980-nm laser also increased ABCA1-mediated cholesterol efflux in THP-1-derived foam cells. This effect was probably mediated by the induction of autophagy through the ROS/PI3K/Akt/mTOR signalling pathway [608,609]. A positive effect of different formulas of Ce6 photosensitiser on atheroma plaques, which may be partially explained by increased cholesterol efflux, was observed also in vivo in mice [610] and rabbits [611]. “
We searched databases with the combination of keywords: photodynamic therapy, photosynthetizers, photosensitizers, atherosclerosis, and HDL. We found that the role of photodynamic therapy in atherosclerosis is to trigger different death pathways (apoptosis, necrosis) in cells occurring in atherosclerotic plaque. Only one connection between photodynamic therapy in atherosclerosis and lipoprotein particle is the use of ox-LDL as a delivery system for photosensitisers to cells with increased uptake of LDL in atherosclerotic lesions. The search with the combination of photodynamic therapy and HDL yielded only the use of HDL as a carrier for photosensitisers in the treatment of cancer. The in vitro and in vivo studies (added to our manuscript) in which increased cholesterol efflux in foam macrophages following the induction of autophagy by photodynamic therapy was observed, was the only connection of photosensitisers to the HDL function.
We also added two paragraphs about the use of antibodies in described research, and changed table 3, figure 2, and graphical abstract.
„Besides synthetic molecules, the activity of CETP may be inhibited also by monoclonal antibodies[480] and the special vaccination approach, eliciting anti-CETP antibodies in order to reduce its activity, was also tested in clinical trials [481-483].“ (page 31, rows 1430-1433)
“Another potential molecule for improving HDL quality is monoclonal antibody MEDI5884. The antibody is targeted against endothelial lipase (EL), the phospholipase responsible for HDL metabolism. EL promotes lipid depletion, destabilization, and renal clearance of HDL particles [506]. In vivo experiments in mice revealed that antibody-mediated inhibition of EL led to increased HDLc and the size of HDL particles [507]. A positive effect of MEDI5884 on HDL was confirmed in non-human primates, but also in a human Phase I study on healthy volunteers, where MEDI5884 was able to increase HDLc, average HDL size, improved cholesterol efflux, and anti-inflammatory properties of HDL [506], and similar encouraging results were seen also in Phase IIa study on patients with stable coronary artery disease [508].” (Page 32, rows 1480-1489)
Point 2: Please use prisma analysis.
Answer 2: Dear reviewer. As our review is very comprehensive, it may appear as a systematic review. Our plan was to write a narrative review. The review covers a wide range of subjects, it is not pointed to one topic. For every topic, we started a new search in PubMed, Scopus, Web of Science, The European Medicines Agency website, and Clinicaltrials.gov website, and used multiple keywords, and sometimes found an interesting study cited in the publication, so it is not possible to involve searching criteria and prisma analysis.
Please, see the attachment.
Sincerely
Authors

Reviewer 2 Report
Review of the manuscript which has been submitted to Pharmaceuticals-Manuscript no. pharmaceuticals-1947274
In the current context of the chronic inflammatory diseases research domain, the study topic of the review entitled “Alterations of HDL´s to piHDL´s proteome in patients with chronic inflammatory diseases, and HDL-targeted therapies” is well chosen and the resulting review is impressive. The data are very detailed and precisely described. Many metabolic mechanisms involved in the manifestation of inflammatory diseases are described in detail. The authors have carried out a very detailed study of the currently existing therapies, as well as their mechanism of action. The bibliography is very well documented and vast, an impressive study of literature! Anyway, below I have made some suggestions to improve the quality of the work in order to recommend acceptance for publication.
· The review is very long; I believe that a content section at the beginning of the review will be well received in order to follow the information more easily.
· Page 8, rows 391-392: Please reformulate the sentence for a better understanding “The results of studies in humans, in transgenic animal models, or in different in vitro models of cell cultures,”;
· Page 8, row 419: use italics for “in-vivo” and “in-vitro” throughout the text;
· Page 16, row 804: Please reformulate the sentence for a better understanding “to stimulate Nod like-receptor protein 3 (NLRP3)-dependent IL-1β secretion in macrophages in vitro”;
Author Response
Dear reviewer,
We would like to thank you for taking the time and effort necessary to review the manuscript. We appreciate your comments and suggestions, which have been helpful in improving the manuscript.
We accepted all your comments and revised our manuscript. Revisions are marked using “Track Changes” to easily view all changes.
Point 1: The review is very long; I believe that a content section at the beginning of the review will be well received in order to follow the information more easily.
Answer 1: Thank you for this comment. The review will be more transparent after this revision. We added a content section, which is at the beginning of the review.
Point 2: Page 8, rows 391-392: Please reformulate the sentence for a better understanding “The results of studies in humans, in transgenic animal models, or in different in vitro models of cell cultures,”;
Answer 2: “The results of studies that evaluate the effect of apoA-II on atherogenesis (in humans, in transgenic animal models, or in different in vitro models) are controversial.” The position of this sentence is now on page 9, rows 436-438.
Point 3: Page 8, row 419: use italics for “in-vivo” and “in-vitro” throughout the text;
Answer 3: Thank you for noting these mistakes. Italics style of all words (in vitro (18x), in vivo (8x), via (33x), Porphyromonas gingivalis, Paeonia suffruticosa, Gymnema Sylvestre, Lactobacillus (L.) casei, L. rhamnosus, L. sporogenes, L. acidophilus (3x), APOE isoform, APOC4 polymorphism, APOJ single nucleotide polymorphism, AAT gene, de novo(2x)) was lost following the submission. We corrected all of them.
Point 4: Page 16, row 804: Please reformulate the sentence for a better understanding “to stimulate Nod like-receptor protein 3 (NLRP3)-dependent IL-1β secretion in macrophages in vitro”;
Answer 3: “In contrast, in vitro studies revealed that the ability of SAA to stimulate IL-1β secretion in macrophages (via Nod-like-receptor protein 3 (NLRP3)) was abolished upon associating SAA to HDL”. The position of this sentence is now on page 18, rows 848-850.
Please, see the attachment.
Sincerely
